# LEARNING BASIC INTERPRETABLE FACTORS FROM TEMPORAL SIGNALS VIA PHYSICAL SYMMETRY

## ABSTRACT

We have recently seen great progress in learning interpretable music representations, ranging from basic factors, such as pitch and timbre, to high-level concepts, such as chord andtexture. However, most methods rely heavily on music domain knowledge and it remains an open question how to learn interpretable and disentangled representations using inductive biases that are more general. In this study, we use *physical symmetry* as a self-consistency constraint on the latent space. Specifically, it requires the prior model that characterises the dynamics of the latent states to be *equivariant* with respect to a certain group transformation. We show that our model can learn *linear* pitch factor (that agrees with human music perception) as well as pitch-timbre disentanglement from unlabelled monophonic music audio. In addition, the same methodology can be applied to computer vision, learning the 3D Cartesian space as well as space-colour disentanglement from a simple moving object shot by a single fixed camera. Furthermore, applying physical symmetry to the prior model naturally leads to *representation augmentation*, a new learning technique which helps improve sample efficiency.

## 1 INTRODUCTION

Interpretable representation-learning models have achieved great progress for various types of time-series data. Taking the *music* domain as an example, tailored deep generative models (Ji et al., 2020) have been developed to learn pitch, timbre, melody contour, chord progression, accompaniment texture, etc. However, most models still rely heavily on domain-specific knowledge. For example, to use pitch scales or instrument labels for learning pitch and timbre representations (Luo et al., 2020; 2019; Engel et al., 2020; Lin et al., 2021; Esling et al., 2018) and to use chords and rhythm labels for learning higher-level representations (Akama, 2019; Yang et al., 2019; Wang et al., 2020; Wei & Xia, 2021). Such an approach is very different from human learning; even without formal music training, one can at least perceive basic factors such as pitch and timbre from the experience of listening to music. In other words, it remains an open question how to learn interpretable music representations using inductive biases that are more general. We see a similar issue in other domains. For instance, various computer-vision models (McCarthy & Ahmed, 2020; Trevithick & Yang, 2021; Mescheder et al., 2019; Riegler et al., 2017) can learn 3D representations of human faces or a particular scene by incorporating domain knowledge (e.g., labelling of meshes and voxels, 3D-specific setups such as multi-cameras, 3D convolution, etc.) but it remains a non-trivial task to trace the 3D location of a simple moving object from monocular videos in a self-supervised fashion.

In this study, we explore to use *physical symmetry* (i.e., symmetry of physical laws) as a weak self-consistency constraint for the learned latent $z$ space. As indicated in Figure 1, this general inductive bias requires that after a certain transformation $S$ (e.g., translation or rotation) in the latent space, the learned prior model $R$, which is the induced physical law describing the temporal flow of the latent states, should output equivariant predictions. Formally, $z_{t+1} = R(z_t)$ if and only if $z_{t+1}^S = R(z_t^S)$, where $z^S = S(z)$. In other words, $R$ and $S$ are commutable operations for $z$, i.e., $R(S(z)) = S(R(z))$. Note that this approach is fundamentally different from most existing symmetry-informed models (Bronstein et al., 2021), in which the symmetry property is used to constrain the encoder or the decoder.

Specifically, we design **s**elf-supervised learning with **p**hysics **s**ymmetry (**SPS**), a method that adopts an encoder-decoder framework and applies physical symmetry to the prior model. We show that

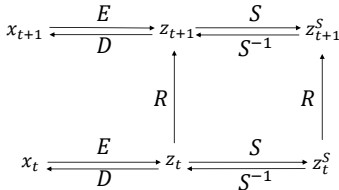

Figure 1: Physical symmetry, the fundamental inductive bias of this study.

with the right symmetry assumptions, our model learns *linear* pitch factors that agree with human music perception from monophonic music audio, without any domain-specific knowledge about pitch scales or signal-level regularities. If we *further* assume an extra global invariant latent code, the model can learn pitch-timbre disentanglement without instrument labelling. Moreover, we show that the same methodology can be applied to the computer vision domain, learning 3D Cartesian space as well as space-colour disentanglement from monocular videos of a bouncing ball shot from a fixed perspective.

## 2 INTUITION

The idea of using physical symmetry for representation learning comes from modern physics. In classical physics, scientists usually first induce physical laws from observations and then discover symmetry properties of the law. (E.g., Newton's law of gravitation, which was induced from planetary orbits, is symmetric with respect to Galilean transformation.) In contrast, in modern physics, scientists often start from a symmetry assumption, based on which they derive the corresponding law and predict the properties (representations) of fundamental particles. (E.g., general relativity was developed based on a firm assumption of symmetry with respect to Lorentz transformation).

Analogously, we use physical symmetry as an inductive bias of our machine learning model, which helps us learn a regularised prior and an interpretable latent space. In other words, if it is a belief of many physicists that symmetry in physical law is a main design principle of the nature, we regard symmetry in physical law as a major useful inductive bias of the representation learner.

The introduction of physical symmetry to the learned prior model naturally leads to **representation augmentation**, a novel learning technique which helps improve sample efficiency. As indicated in Figure 1, representation augmentation means to "imagine" $z_t^S$ as the training sample of the prior model $R$. Representation augmentation can be regarded as a regularisation of the prior model, since it requires the prediction of the $z$ sequence to be *equivariant* with respect to a certain group transformation of $S$. It also constrains the encoder and decoder *indirectly through the prior model* since the network is trained in an end-to-end fashion.

## 3 METHODOLOGY

Our goal is to learn a disentangled and interpretable representation $z_i$ of each high-dimensional sample $x_i$ from time-series $\mathbf{x}_{1:T}$. **The disentanglement of $z_i$ is at two levels**. **First**, $z_i$ is divided into two factors: $z_{i,s}$ and $z_{i,c}$, where $z_{i,s}$ is the global invariant style and $z_{i,c}$ is the content representation that changes over time. **More importantly**, we aim to further disentangle the spatio-temporal content factor $z_{i,c}$ using physical symmetry such that it is equivariant with respect to the prior model and *each* dimension of it is interpretable and consistent with human perception.

We focus on two specific problems in this paper. The primary problem is to learn *pitch* and *timbre* factors of music notes from music audio, where each $x_i$ is a spectrogram of a note. Ideally, $z_{i,c}$ is a 1D content factor representing the pitch and $z_{i,s}$ is a style factor representing the timbre. Another problem is to learn *3D Cartesian location* and *colour* factors of a simple moving object (a bouncing ball) from its trajectory shot by a fixed, single camera. In this case, each $x_i$ is an image. Ideally, $z_{i,c}$ is learned to be a 3D content factor representing the location and $z_{i,s}$ represents the global colour.

### 3.1 MODEL

Figure 2 shows our model design. During the training process, the temporal data input $\mathbf{x}_{1:T}$ is first fed into the encoder $E$ to obtain the corresponding representation $\mathbf{z}_{1:T}$. $\mathbf{z}_{1:T}$ is then split into two parts: the style factor $\mathbf{z}_{1:T,s}$ and the content factor $\mathbf{z}_{1:T,c}$. The style factor $\mathbf{z}_{1:T,s}$ is passed through the random-pooling module $P$, where one sub-element $z_{\tau,s}$ is randomly picked. The content factor $\mathbf{z}_{1:T,c}$ is fed into *three* branches. In the first branch (the green line), $\mathbf{z}_{1:T,c}$ is combined with $z_{\tau,s}$ by concatenating each sub-element $z_{i,c}$ with $z_{\tau,s}$, and fed to the decoder $D$ to reconstruct $\mathbf{x}'_{1:T}$. In the second branch (the orange line), $\mathbf{z}_{1:T,c}$ is passed through the prior model $R$ to predict its next step, $\hat{\mathbf{z}}_{2:T+1,c}$, which is then combined with $z_{\tau,s}$ to reconstruct $\hat{\mathbf{x}}_{2:T+1}$. In the third branch (the blue line), we sequentially transform $\mathbf{z}_{1:T,c}$ with $S$, pass it through $R$, and transform it back using the inverse transformation $S^{-1}$ to predict another version of the next step $\tilde{\mathbf{z}}_{2:T+1,c}$, which is finally combined with $z_{\tau,s}$ to reconstruct $\tilde{\mathbf{x}}_{2:T+1}$. We get three outputs from the model: $\mathbf{x}'_{1:T}$, $\hat{\mathbf{x}}_{2:T+1}$ and $\tilde{\mathbf{x}}_{2:T+1}$.

The underlying idea of physical symmetry is that the dynamics of latent content factor and its transformed version *follow the same physical law* characterised by $R$. Therefore, $\tilde{\mathbf{z}}$ and $\hat{\mathbf{z}}$ should be close to each other and so are $\tilde{\mathbf{x}}$ and $\hat{\mathbf{x}}$, assuming $S$ is a proper transformation. This self-consistency constraint helps the network learn a more regularised latent space. In addition, $z_{\tau,s}$ further helps disentangle the style factor by assuming a global invariant style code over time.

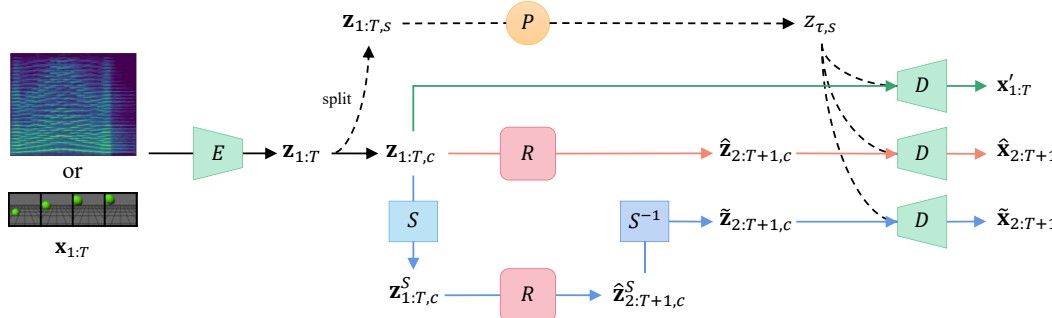

Figure 2: An overview of our model. $\mathbf{x}_{1:T}$ is fed into the encoder $E$ to obtain the corresponding representation $\mathbf{z}_{1:T}$, which is then split into two parts: the style factor $\mathbf{z}_{1:T,s}$ and the content factor $\mathbf{z}_{1:T,c}$. The style factor is passed through the random-pooling layer $P$, where a sub-element $z_{\tau,s}$ is randomly selected. The content factor is fed into three different branches and combined with $z_{\tau,s}$ to reconstruct three outputs respectively: $\mathbf{x}'_{1:T}$, $\hat{\mathbf{x}}_{2:T+1}$ and $\tilde{\mathbf{x}}_{2:T+1}$. Here, $R$ is the prior model and $S$ is the symmetric operation. The inductive bias of physical symmetry enforces $\mathbf{z}_{1:T,c}$ to be equivaraint w.r.t. to $R$, so therefore $\tilde{\mathbf{z}}$ and $\hat{\mathbf{z}}$ should be close to each other and so are $\tilde{\mathbf{x}}$ and $\hat{\mathbf{x}}$.

### 3.2 TRAINING OBJECTIVE

The total loss contains four terms: reconstruction loss $\mathcal{L}_{\text{rec}}$, prior prediction loss $\mathcal{L}_{\text{prior}}$, symmetry-based loss $\mathcal{L}_{\text{sym}}$, and KL divergence loss $\mathcal{L}_{\text{KLD}}$. Formally,

$$\mathcal{L} = \mathcal{L}_{\text{rec}} + \lambda_1 \mathcal{L}_{\text{prior}} + \lambda_2 \mathcal{L}_{\text{sym}} + \lambda_3 \mathcal{L}_{\text{KLD}}, \tag{1}$$

where $\lambda_1$, $\lambda_2$ and $\lambda_3$ are weighting parameters. By referring to the notations in section 3.1,

$$\mathcal{L}_{\text{rec}} = \mathcal{L}_{\text{BCE}}(\mathbf{x}'_{1:T}, \mathbf{x}_{1:T}) + \mathcal{L}_{\text{BCE}}(\hat{\mathbf{x}}_{2:T}, \mathbf{x}_{2:T}) + \mathcal{L}_{\text{BCE}}(\tilde{\mathbf{x}}_{2:T}, \mathbf{x}_{2:T}), \tag{2}$$

$$\mathcal{L}_{\text{prior}} = \ell_2(\hat{\mathbf{z}}_{2:T,c}, \mathbf{z}_{2:T,c}), \tag{3}$$

$$\mathcal{L}_{\text{sym}} = \ell_2(\tilde{\mathbf{z}}_{2:T,c}, \hat{\mathbf{z}}_{2:T,c}) + \ell_2(\tilde{\mathbf{z}}_{2:T,c}, \mathbf{z}_{2:T,c}). \tag{4}$$

Lastly, the $\mathcal{L}_{\text{KLD}}$ is the Kulback-Leibler divergence loss between the posterior distribution of $z_i$ and a standard Gaussian.

### 3.3 Symmetry-based representation augmentation

We name $S$ with *representation augmentation* since it creates extra fake sequences of $z$ (i.e., imaginary experience based on a group assumption) to help train the prior. In practice, we apply $K$ different transformations $S$ to $\mathbf{z}_{1:T-1,c}$ to generate $K$ fake sequences. Thus, the two terms of symmetry-based loss can be specified as:

$$\ell_2(\tilde{\mathbf{z}}_{2:T,c}, \mathbf{z}_{2:T,c}) = \frac{1}{K}\sum_{k=1}^{K}\ell_2(S_k^{-1}(R(S_k(\mathbf{z}_{1:T-1,c}))), \mathbf{z}_{2:T,c}), \tag{5}$$

$$\ell_2(\tilde{\mathbf{z}}_{2:T,c}, \hat{\mathbf{z}}_{2:T,c}) = \frac{1}{K}\sum_{k=1}^{K}\ell_2(S_k^{-1}(R(S_k(\mathbf{z}_{1:T-1,c}))), \hat{\mathbf{z}}_{2:T,c}), \tag{6}$$

where the lower case $k$ denotes the index of a specific transformation and we refer to $K$ as the *augmentation factor*. Likewise, the last term of reconstruction loss can be specified as:

$$\mathcal{L}_{\text{BCE}}(\tilde{\mathbf{x}}_{2:T}, \mathbf{x}_{2:T}) = \frac{1}{K}\sum_{k=1}^{K}\mathcal{L}_{\text{BCE}}(D(S_k^{-1}(R(S_k(\mathbf{z}_{1:T-1,c}))), z_{\tau,s}), \mathbf{x}_{2:T}). \tag{7}$$

Each $S$ applied to each sequence $\mathbf{z}_{1:T,c}$ belongs to a certain group, and different groups are used for different problems. For the music problem, we assume $\mathbf{z}_{i,c}$ be to 1D and use random $S \in G \cong (\mathbb{R}, +)$. In other words, we add or subtract the content factor by a random scalar. As for the video problem, we assume $\mathbf{z}_{i,c}$ be to 3D and use random $S \in G \cong (\mathbb{R}^2, +) \times \text{SO}(2)$. In other words, random rotation and shift are applied on two dimensions of $\mathbf{z}_{i,c}$.

## 4 Results

We run experiments on two problems. In section 4.1, we present the results of learning pitch content and timbre style from monophonic music. In section 4.2, we present the results of learning 3D-location content and colour style from monocular videos of a bouncing ball. For both problems, we use a representation augmentation factor of $K = 4$. The focus in this section is to see whether the learned content is interpretable, and whether the content-style disentanglement is successful. We present additional results in the appendix, including more complicated and realistic data in section A.3, more flexible assumptions on symmetry in section A.2.4, and results of applying SPS on auto-encoders without the variational constraints in section A.4

### 4.1 Learning pitch and timbre factors from music audio

#### 4.1.1 Dataset

We synthesise a dataset that contains around 2400 audio clips played by multiple instruments. Each contains 15 notes in major scales with the first half ascending and the second half descending. We refer readers to appendix section A.1.3 for details.

#### 4.1.2 Results on interpretable pitch space

Figure 3 shows that the pitch factor learned by our model has a linear relation with the true pitch. Here, we use $z_{\text{pitch}}$ as the synonym of $z_c$ to denote the content factor. The plot shows the mappings of two tasks and four models. In the embedding task (the first row), $x$-axis is the true pitch and $y$-axis is embedded $z_{\text{pitch}}$. In the synthesis task (the second row), $x$-axis is $z_{\text{pitch}}$ and $y$-axis is the detected pitch (by YIN algorithm, a standard pitch-estimation method by De Cheveigné & Kawahara (2002)) of decoded (synthesised) notes. The fours models involved are: 1) our model, 2) our model without symmetry (i.e., no representation augmentation during training), 3) a $\beta$-VAE trained to encode single-note spectrograms from a single instrument (banjo) to 1D embeddings, and 4) SPICE (Gfeller et al., 2020), a SOTA unsupervised pitch estimator *with strong domain knowledge on how pitch linearity is reflected in log-frequency spectrograms*. As the figure shows, without explicit knowledge of pitch, our model learns a more interpretable pitch factor than $\beta$-VAE, and the result is comparable to SPICE.

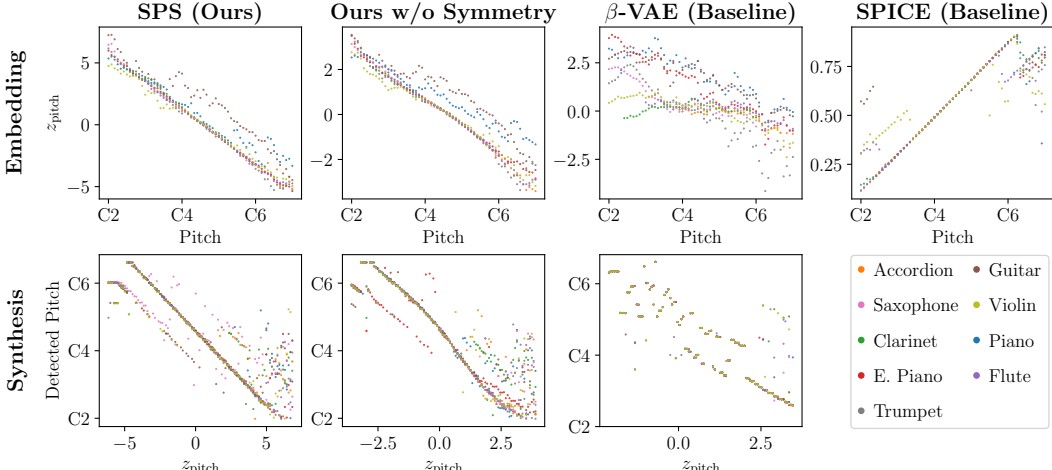

Figure 3: A visualisation of the mapping between the 1D content factor and the true pitch. In the upper row, models encode notes in the test set to $z_{\text{pitch}}$. The $x$ axis shows the true pitch and the $y$ axis shows the learned pitch factor. In the lower row, the $x$ axis traverses the $z_{\text{pitch}}$ space. The models decode $z_{\text{pitch}}$ to audio clips. We apply YIN to the audio clips to detect the pitch, which is shown by the $y$ axis. In both rows, a linear, noiseless mapping is ideal, and our method performs the best. All results are evaluated on the test set.

### 4.1.3 RESULTS ON PITCH-TIMBRE DISENTANGLEMENT

We evaluate the content-style disentanglement using factor-wise data augmentation following Yang et al. (2019). Namely, we change (i.e., augment) the instrument (i.e., style) of notes while keeping their pitch, and then measure the effects on the encoded $z_c$ and $z_s$. We compare the normalised $z_c$ and $z_s$, ensuring they have the same dynamic range. Ideally, the change of $z_s$ should be much more significant than $z_c$. Here, we compare four approaches: 1) our model, 2) our model without splitting for $z_s$ (i.e., our model without the dashed black path shown in Figure 2) but fed $z_s$ together with $z_c$ into the prior model, 3) GMVAE (Luo et al., 2019), a domain-specific pitch-timbre disentanglement model trained with *explicit pitch labels*, and 4) TS-DSAE (Luo et al., 2022), a latest unsupervised pitch-timbre disentanglement model based on Disentangled Sequential Autoencoder (DSAE).

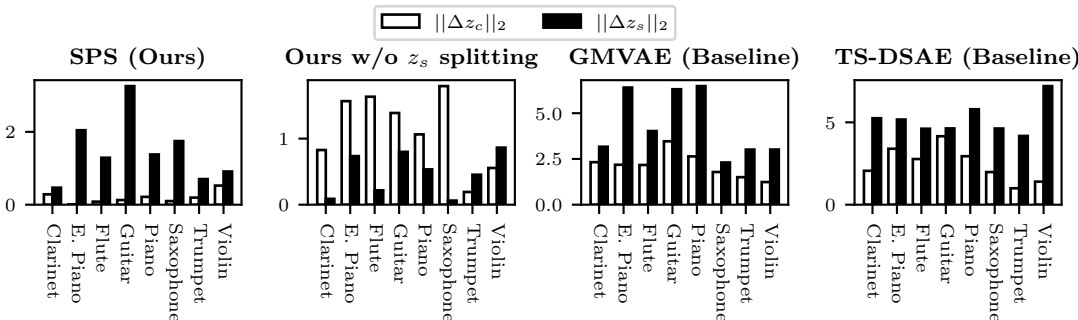

Figure 4: Comparisons for $\Delta z_c$ and $\Delta z_s$ for different instruments against accordion, with pitch kept constant at MIDI pitch D3. $\Delta z_c$ and $\Delta z_s$ are changes in normalised $z_c$ and $z_s$, so that higher black bars relative to white bars means better results. All results are evaluated on the test set.

Figure 4 presents the changes in normalised $z_c$ and $z_s$ measured by L2 distance when we change the instrument of an anchor note whose pitch is D3 and synthesised by accordion. Table 1 provides a more quantitative version by aggregating all possible instrument combinations and all different pitch pairs. Both results show that our model produces a smaller relative change in $z_c$ under timbre

augmentation, demonstrating a successful pitch-timbre disentanglement outperforming both the ablation and baseline. Note that for the ablation model, $z_c$ varies heavily under timbre augmentation, seemingly containing timbre information. This result indicates that the design of an invariant style factor over the temporal flow is necessary to achieve good disentanglement.

Table 1: Mean ratios of changes in normalised $z_c$ and $z_s$ under timbre augmentation across all possible instrument combinations under different constant pitches in the test set.

| Methods | $||\Delta z_c||_2/||\Delta z_s||_2 \downarrow$ |
|---|---|
| Ours w/o $z_s$ splitting | 2.20 |
| GMVAE (Baseline) | 0.67 |
| DS-DSAE (Baseline) | 0.65 |
| SPS (Ours) | **0.49** |

## 4.2 LEARNING LOCATION AND COLOUR FACTORS FROM VIDEOS OF A MOVING OBJECT

### 4.2.1 DATASET

We run physical simulations of a bouncing ball in a 3D space and generate 4096 trajectories. The ball's colour, initial location, and initial velocity for each trajectory are randomly sampled. Please see appendix A.1.3 for more details.

### 4.2.2 RESULT ON INTERPRETABLE 3D REPRESENTATION

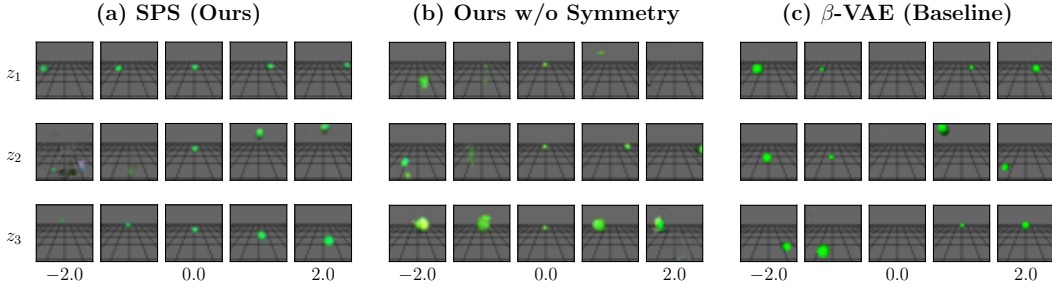

Figure 5: Row $i$ shows the generated images when changing $z_i$ and keeping $z_{\neq i} = 0$, where the $x$ axis varies $z_i$ from $-2$ to $2$. In (a), changing $z_2$ controls the ball's height, and changing $z_1, z_3$ moves the ball parallel to the ground plane.

Figure 5 illustrates the interpretability of learned content factor using latent space traversal. Each row varies only one dimension of the learned 3D content factor, keeping the other two dimensions at zero. Figure 5(a) shows the results of our model. We clearly observe that: i) *increasing $z_1$ (the first dimension of $z_c$) mostly moves the ball from left to right, increasing $z_2$ moves the ball from bottom to top, and increasing $z_3$ mostly moves the ball from far to near.* Figure 5(b) is the ablation model without physical symmetry, and (c) shows the result of our baseline model $\beta$-VAE, which is trained to reconstruct static images of a single colour (green). Neither (b) nor (c) learns an interpretable latent space.

Table 2 quantitatively evaluates the linearity of the learned location factor. We fit a linear regression from $z_c$ to the true 3D location over the test set and then compute the Mean Square Errors (MSEs). A smaller MSE indicates a better fit. All three methods (as used in Figure 5) are evaluated on a single-colour (green) test set. Results show that our model achieves the best linearity in the learned latent factors, which aligns with our observations in Figure 5.

Table 2: Linear fits between the true location and the learned location factor. We run the encoder on the test set to obtain data pairs in the form of (location factor, true coordinates). We then run a linear fit on the data pairs to evaluate factor interpretability.

| Method | $x$ axis MSE ↓ | $y$ axis MSE ↓ | $z$ axis MSE ↓ | MSE ↓ |
|---|---|---|---|---|
| $\beta$-VAE (Baseline) | 0.37 | 0.76 | 0.73 | 0.62 |
| Ours w/o Symmetry | 0.35 | 0.72 | 0.68 | 0.58 |
| SPS (Ours) | **0.11** | **0.06** | **0.09** | **0.09** |

### 4.2.3 RESULT ON SPACE-COLOUR DISENTANGLEMENT

Similar to section 4.1.3, we evaluate the space-colour disentanglement by augmenting the colour (i.e., style) of the bouncing balls while keeping their locations, and then measure the effects on the normalised $z_c$ and $z_s$. Again, a good disentanglement should lead to a change in $z_s$ much more significant than $z_c$. Here, we compare two approaches: 1) our model and 2) our model ablating splitting for $z_s$. The ablation model follows the same configuration as in section 4.1.3. A subtle difference is that the ablation model does not differently constrain $z_2$ (corresponding to the $y$-axis) than $z_s$. To ensure a meaningful comparison, under colour augmentation, we consider $z_2$ to be a part of $z_s$ of the ablation model and a part of $z_c$ of the complete model.

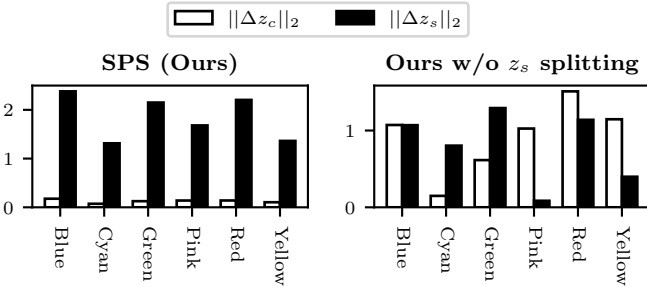

Figure 6: Comparisons of normalised $\Delta z_c$ and $\Delta z_s$ for different colours against white, with the ball's location kept constant at (0, 1, 5). Higher black bars (relative white bars) means better a result. (Results are evaluated on the test set.)

Table 3: Mean ratios of changes in normalised $z_c$ and $z_s$ under colour augmentation across sampled colour combinations keeping locations constant. Results are evaluated on the test set.

| Methods | $||\Delta z_c||_2/||\Delta z_s||_2$ ↓ |
|---|---|
| Ours w/o $z_s$ splitting | 1.62 |
| SPS (Ours) | **0.54** |

Figure 6 presents the changes in normalised $z_c$ and $z_s$ measured by L2 distance when we change the instrument of an anchor ball whose location is (0, 1, 5) and rendered using white colour. Table 3 provides a more quantitative version by aggregating sampled colour combinations and location pairs. Both results show that our model produces a smaller relative change in $z_c$ under timbre augmentation, demonstrating a successful pitch-timbre disentanglement outperforming the ablation model. Note that for the ablation model, $z_c$ varies heavily under colour augmentation. This result agrees with section 4.1.3 and again indicates that the design of an invariant style factor helps with disentanglement.

## 5 ANALYSIS

To better understand how representation augmentation leads to an interpretable content factor, we train a simplified version of our model with no style factor on the videos of a single-colour (green) bouncing ball. We choose the vision problem since a 3D content manifests a more obvious difference when physical symmetry is applied.

### 5.1 REPRESENTATION AUGMENTATION IMPROVES SAMPLE EFFICIENCY

Figure 7 shows that *a larger factor of representation augmentation leads to lower linear projection loss* (the measurement introduced in section 4.2.2) of the learned 3D representation. Here, $K$ is the augmentation factor, and $K = 0$ means the model is trained without physical symmetry. The comparative study is conducted on 4 training set sizes (256, 512, 1024 and 2048), in which each box plot shows the results of 10 experiments trained with a fixed $K$ and random initialisation. We see that a larger $K$ leads to better results and compensates for the lack of training data. E.g., the loss trained on 256 samples with $K = 4$ is comparable to the loss trained on 1024 samples with $K = 0$, and the loss trained on 512 samples with $K = 4$ is even lower than the loss trained on 2048 samples with $K = 0$. In other words, when $K = 0$, increasing the number of training samples beyond a certain point does not further shrink the error, but increasing $K$ still helps. In appendix A.2.4, we further show that even with *incorrect* group assumptions, our method still helps learn interpretable content factors.

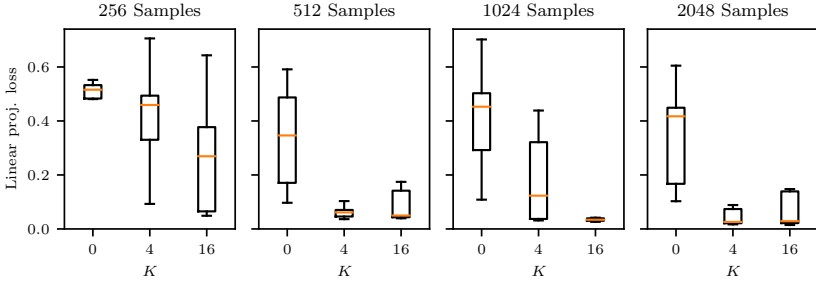

Figure 7: Linear projection MSE for different augmentation factors ($K$) and training set sizes. Representation augmentation improves sample efficiency as smaller values mean better results.

### 5.2 REPRESENTATION AUGMENTATION IMPROVES DISENTANGLEMENT

Figure 8 visualises the latent space during different stages of model training, and we see that a larger $K$ leads to a better disentanglement in an earlier stage of training. The horizontal axis shows the training epoch. Three experiments with different $K$ values ($\times 0, \times 4, \times 16$) are stacked vertically. Each experiment is trained twice with random initialisation. Each subplot shows the orthogonal projection of the $z_c$ space onto the plane spanned by $z_1$ and $z_3$, therefore hiding most of the $y$-axis (i.e. ball height) wherever disentanglement is successful. During training, the role of physical symmetry is to "straighten" the encoded grid and a larger $K$ yields a stronger effect.

## 6 RELATED WORK

The idea of using a predictive model for better self-supervised learning has been well established (Oord et al., 2018; Chung et al., 2015; LeCun, 2022). In terms of model architecture, our model belongs to the family of disentangled sequential autoencoders (Bai et al., 2021; Hsu et al., 2017; Vowels et al., 2021; Yingzhen & Mandt, 2018; Zhu et al., 2020). It is very similar to VRNN (Chung et al., 2015) if we remove the global invariant part from the overall network. In addition, our model can be seen as a variation of joint-embedding predictive architecture (JEPA) in LeCun (2022) if we eliminate the reconstruction losses on the observation. In fact, we see the network topology of a model as the "hardware" and see the learning strategy (e.g., contrastive method, regularised method,

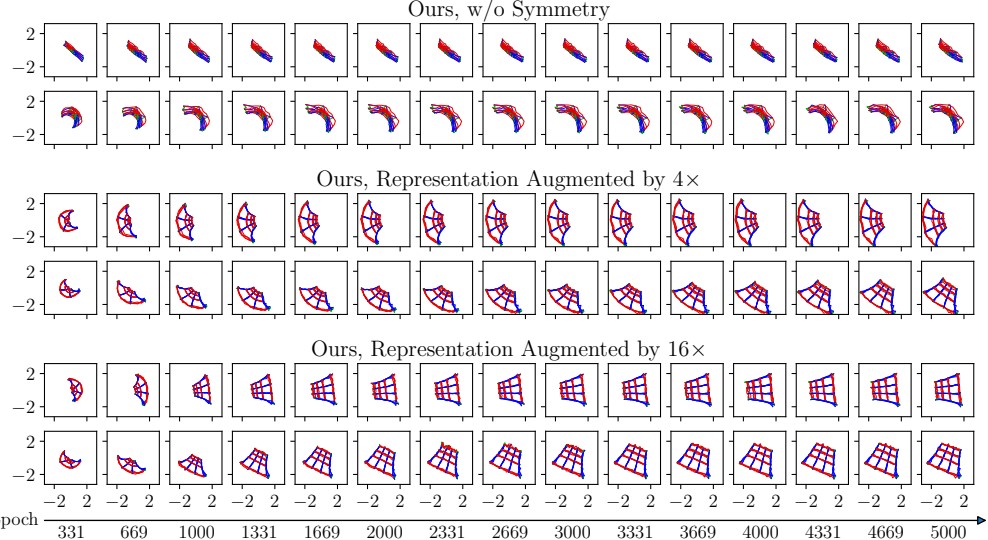

Figure 8: A visualisation of the learned latent space against training epoch. We plot how the encoder projects an equidistant 3D grid of true Cartesian coordinates onto the $z$ space. Different colours denote respective axes in the true coordinates.

or a mixed one), as "software". The main contribution of this study lies in the learning strategy — to use physical symmetry to limit the capacity of the latent space, and to use representation augmentation to increase sample efficiency.

The notion of "symmetry" and the use of symmetry for disentanglement are also not new in representation learning (Higgins et al., 2018; Bronstein et al., 2021). Most symmetric-based methods care about the relation between observation $x$ and latent $z$ (Sanghi, 2020; Quessard et al., 2020; Dupont et al., 2020; Huang et al., 2021). E.g., when a certain transformation is applied to $x$, $z$ should simply keep invariant or follow a same/similar transformation. Such an assumption inevitably requires some knowledge in the domain of $x$. In contrast, the physical symmetry used in this study focuses solely on the dynamics of $z$, and therefore we only have to make assumptions about the underlying group transformation in the latent space. We see two most relevant works in the field of reinforcement learning (Mondal et al., 2022; Dupont et al., 2020), which applied an equivariant assumption very similar to the physical symmetry used in this paper. The major differences are twofold. First, to disentangle the basic factors, our method requires no interactions with the environment. Second, our method is much more concise; it needs no other tailored components or other inductive biases such as symmetric embeddings network and contrastive loss used in Dupont et al. (2020) or MDP homomorphism applied in Mondal et al. (2022).

## 7 CONCLUSION

In this paper, we propose a methodology that uses physical symmetry to learn disentangled and interpretable representations from time-series data. Experiments show that physical symmetry effectively regularises the latent space by enforcing the prior model to be equivariant with respect to group transformations. Under a proper group assumption, our method learns an interpretable 1D pitch factor (that agrees with human music perception) as well as a good pitch-timbre disentanglement from music audios without any labels of pitches or instruments. We also show that with the same method, we can learn an interpretable 3D Cartesian location factor as well as a good location-colour disentanglement from monocular videos of bouncing ball shot from a fixed perspective. In addition, a new training technique, representation augmentation, is developed to couple with physical symmetry. Analysis shows that representation augmentation leads to better sample efficiency and representation disentanglement.

## REPRODUCIBILITY STATEMENT

The source code for training and testing the models, as well as generating the figures and tables, is publicly available at https://github.com/double-blind-75098/Learning-basic-interpretable-factors-from-temporal-signals-via-physical-symmetry.

## ACKNOWLEDGMENTS

Redacted for review.

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

## A    APPENDIX

In A.1, we present implementation details of architecture, training process, and datasets. In A.2, we present extra experimental results on synthetic data, including data reconstruction, more evaluation on disentanglement, and more analysis on physical symmetry and representation augmentation. In A.3 we show how our model performs on more realistic and complicated problems. Finally, we show an autoencoder version of our method in A.4.

## A.1 Implementation details

### A.1.1 Architecture details

Out models for both tasks share the following architecture. The encoder is composed of a 2D-CNN with ReLU activation connected by two parallel linear layers, one for estimating the mean and the other for the log-variance. The prior model is a vanilla RNN of one layer with 256 hidden units and one linear layer projection head. The decoder consists of several linear layers followed by a set of 2D transposed convolution layers mirroring the CNN in the encoder. We apply the Sigmoid function to its output to produce the final output. We use no batch normalisation or dropout layers.

Minor variations exist between the models for the two tasks. In the audio task, we use three convolution layers in the encoder, with three linear and three 2D transposed convolution layers in the decoder. In the vision task, as the data is more complex, we use four convolution layers in the encoder, with four linear and four 2D transposed convolution layers in the decoder.

### A.1.2 Training details

For both tasks, we use the Adam optimiser with a learning rate gradually decreasing from $10^{-3}$ to $10^{-5}$ in about 150000 batch iterations. The training batch size is 32 across all of our experiments. For all VAE-based models we present, including SPS (ours), ours w/o symmetry, and $\beta$-VAE (baseline), we set $\beta$ (i.e., $\lambda_3$ in Equation (1)) to 0.01, with $\lambda_1 = 1$ and $\lambda_2 = 2$. All BCE and MSE loss functions are calculated in sum instead of mean. $K = 4$ for all SPS models except for those discussed in section 5 where we analyse the influence of different $K$.

The RNN predicts $z_{n+1:T,c}$ given the first $n$ embeddings $z_{1:n,c}$. We choose $n = 3$ for the audio task and $n = 5$ for the vision task. We adopt scheduled sampling (Bengio et al., 2015) during the training stage, where we gradually reduce the guidance from teacher forcing. After around 50000 batch iterations, the RNN relies solely on the given $z_{n+1:T,c}$ and its predictions.

For random pooling in the training stage, one style vector is randomly selected from all time steps (i.e., 15 for the music task and 20 for the vision task) of the sequence to represent $z_s$. In the testing stage, only the first 5 (vision task) or 3 (music task) frames are given, and $z_s$ will be selected from them.

**Audio-specific setups**. We run STFT (with sample rate $= 16000/s$, window length $= 1024$, hop length $= 512$, and no padding) over each audio to obtain a spectrogram. We further convert its energy values into a logarithmic scale and normalise them to the range $[0, 1]$. We then slice the spectrogram into fifteen segments, each containing one note. The CNN encoder, in each timestep, takes one segment as input. In practice, we sample $S \sim \mathcal{U}([-1, 1])$. We set the dimension of $z_c$ to 1 and the dimension of $z_s$ to 2.

**Vision-specific setups**. In practice, we randomly choose from $S_1 \in G_1 \cong (\mathbb{R}^2, +)$ and $S_2 \in G_2 \cong \mathrm{SO}(2)$ to augment the representation. Both of them apply to two dimensions of $z_c$. Under the constraints of symmetry these two dimensions represent the horizontal $x$-$z$ plane. Similar to the audio setup, we sample $S_1 \sim \mathcal{U}([-1, 1]^2)$. We set the dimension of $z_c$ to 3 and the dimension of $z_s$ to 2.

### A.1.3 Dataset details

**Music problem** For the synthesized music scale dataset, each clips each containing 15 notes in major scales with the first half ascending and the second half descending. The tone of each note has the same volume and duration. The interval between every two notes is equal. We vary the starting pitch such that every MIDI pitch in the range C2 to C7 is present in the dataset. For each note sequence, we synthesise it using 53 different instruments, yielding 2376 audio clips. Specifically, two soundfonts are used to render those audio clips respectively: FluidR3_GM (Wen, 2013) for the train set and GeneralUser GS v1.471 (Chris, 2017) for the test set. The pitch ranges for different instruments vary, so we limit each instrument to its common pitch range (See Table 15).

**Vision problem** The simulated ball is affected by gravity and bouncing force (elastic force). A fixed camera records a 20-frame video of each 4-second simulation to obtain one trajectory (see Figure 9).

The ball's size, gravity, and proportion of energy loss per bounce are constant across all trajectories.

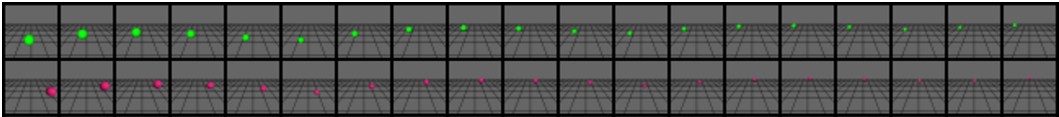

Figure 9: Two example trajectories from the bouncing ball dataset.

## A.2 EXTRA RESULTS

### A.2.1 MORE INSPECTIONS ON LATENT FACTORS

To supplement results presented in section 4.1.2, Figure 10 shows a more quantitative analysis, using $R^2$ as the metric to evaluate the linearity of the pitch against $z_{\text{pitch}}$ mapping. Although SPICE produces rather linear mappings in Figure 3, it suffers from octave errors towards extreme pitches, hurting its $R^2$ performance.

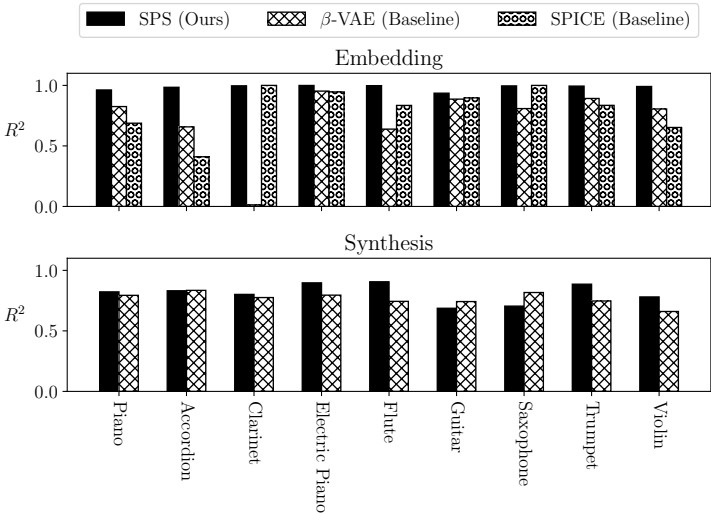

Figure 10: We use $R^2$ to evaluate mapping linearity. A larger $R^2$ indicates a more interpretable latent space. Results are evaluated on the test set.

Figure 11 evaluates the learned colour factor of our model. Each pixel shows the colour of the ball synthesised by the decoder using different $z$ coordinates. The ball colour is detected using naive saturation maxima. In the central subplot, the location factor $z_{1:3}$ stays at zeros while the colour factor $z_{4:5}$ is controlled by the subplot's $x, y$ axes. As shown in the central subplot, our model (a) learns a natural 2D colour space. The surrounding subplots keep the colour factor $z_{4:5}$ unchanged, and the location factor $z_{1,3}$ is controlled by the subplot's $x, y$ axes. A black cross marks the point where the entire $z_{1:5}$ is equal to the corresponding black cross in the central subplot. As is shown by the surrounding subplots, varying the location factor does not affect the colour produced by our model (a), so the disentanglement is successful. The luminosity changes because the scene is lit by a point light source, making the ball location affect the surface shadow. On the other hand, $\beta$-VAE (b) learns an uninterpretable colour factor.

### A.2.2 MORE ON CONTENT-STYLE DISENTANGLEMENT

To supplement sections 4.1.3 and 4.2.3, we further quantify the results in the form of augmentation-based queries following Yang et al. (2019), regarding the intended split in $z$ as ground truth and the

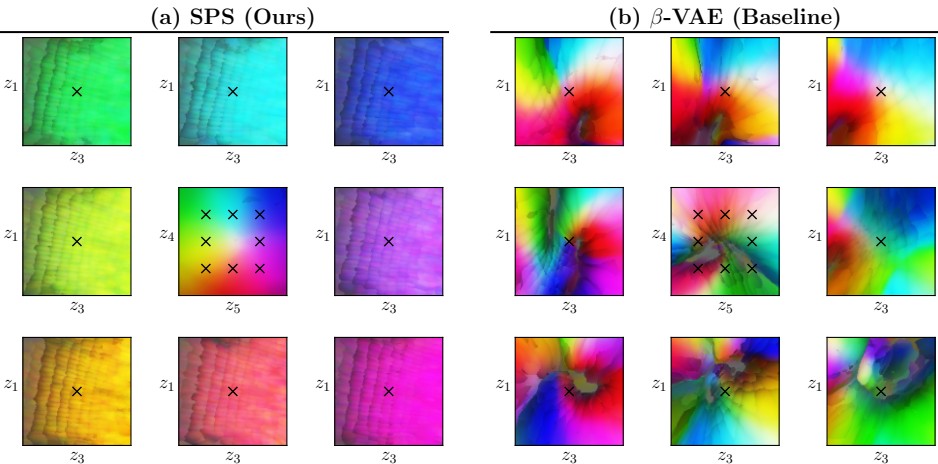

Figure 11: The colour map of the synthesised ball experiment through latent space traversal. Each pixel represents the detected colour from one synthesised image of the ball. Each subplot varies two dimensions of $z$, showing how the synthesised colour responds to the controlled $z$.

dimensions with the largest variances from factor-wise augmentation after normalisation as predictions. For example, under timbre augmentation under a given pitch for our model, if $z_1$ and $z_3$ are the two dimensions of $z$ that produce the largest variances after normalisation, we count one false positive ($z_1$), one false negative ($z_2$), and one true positive ($z_3$). The precision would be 0.67. Tables 4 and 5 show the precision scores of the three approaches against their corresponding random selection for the two tasks. The results are in line with our observation in the previous sections, with our model more likely to produce the largest changes in dimensions in $z_c$ under content augmentation and that in $z_s$ under style augmentation.

Table 4: Results on augmentation-based queries on the audio task. Precision, recall and F1 are the same since the number of predicted and ground-truth positives are identical. Note that random precisions for different approaches can be different as $z_c$ and $z_s$ are split differently.

| Methods | Timbre augmentation | | Pitch augmentation | |
|---|---|---|---|---|
| | Precision ↑ | Random Precision | Precision ↑ | Random Precision |
| Ours w/o $z_s$ splitting | 0.50 | 0.67 | 0.02 | 0.33 |
| GMVAE (Baseline) | 0.93 | 0.50 | 0.83 | 0.50 |
| TS-DSAE (Baseline) | 0.81 | 0.50 | 0.68 | 0.50 |
| SPS (Ours) | 0.98 | 0.67 | 0.82 | 0.33 |

Table 5: Results on augmentation-based queries on the visual task. Since the ablation model does not differently constrain $z_2$ (corresponding to the $y$-axis) than $z_s$, we consider $z_c$ and $z_s$ differently for the two approaches. Under colour augmentation, we consider $z_2$ to be a part of $z_s$ for the ablation model and a part of $z_c$ for the complete model. Under location augmentation, we consider $z_2$ to be a part of $z_c$ for both models.

| Methods | Colour augmentation | | Location augmentation | |
|---|---|---|---|---|
| | Precision ↑ | Random Precision | Precision ↑ | Random Precision |
| Ours, w/o $z_s$ splitting | 0.64 | 0.60 | 0.36 | 0.40 |
| SPS (Ours) | 0.99 | 0.40 | 0.88 | 0.40 |

### A.2.3 RECONSTRUCTION AND PRIOR PREDICTION

We investigate the video reconstruction and prediction capacities of our model and show that they are not harmed by adding symmetry constraints. We compare our model, our model ablating symmetry constraints, and a $\beta$-VAE trained solely for only image reconstruction. Table 6 and Table 7 respectively show the results of audio and vision tasks. For each task, we report per-pixel BCE of the reconstructed sequences from the original input frames (Self-recon) and from the RNN predictions (Pred-recon). We also include $\mathcal{L}_{\mathrm{prior}}$, the MSE loss on the RNN-predicted $\hat{z}$ as defined in section 3.2. The results show that our model suffers little decrease in reconstruction and prediction performance while surpassing the ablation model in terms of $\mathcal{L}_{\mathrm{prior}}$.

Table 6: Reconstruction and prediction results of models in section 4.1.2 on the audio task.

| Methods | Self-recon (BCE/Pixel) ↓ | Pred-recon (BCE/Pixel) ↓ | $\mathcal{L}_{\mathrm{prior}}$ (MSE/$z$) ↓ |
|---|---|---|---|
| Ours w/o Symmetry | 0.0360 | 0.0363 | 0.0486 |
| $\beta$-VAE (Baseline) | 0.0359 | N/A | N/A |
| SPS (Ours) | 0.0356 | 0.0359 | 0.0418 |

Table 7: Reconstruction and prediction results of models in section 4.2.2 on the video task.

| Methods | Self-recon (BCE/Pixel) ↓ | Pred-recon (BCE/Pixel) ↓ | $\mathcal{L}_{\mathrm{prior}}$ (MSE/$z$) ↓ |
|---|---|---|---|
| Ours w/o Symmetry | 0.6456 | 0.6464 | 0.132 |
| $\beta$-VAE (Baseline) | 0.6455 | N/A | N/A |
| SPS (Ours) | 0.6457 | 0.6464 | 0.0957 |

### A.2.4 MORE ANALYSIS ON REPRESENTATION AUGMENTATION

To supplement the results in section 5, Figure 12 shows how representation augmentation jointly affect the linearity of the learned latent space and the reconstruction accuracy of $\hat{x}$. Similar to Figure 7, the experiment is conducted using different $K$s, each with 10 trails. We see that representation augmentation does not sacrifice reconstruction but shrinks the linear projection loss in a consistent manner.

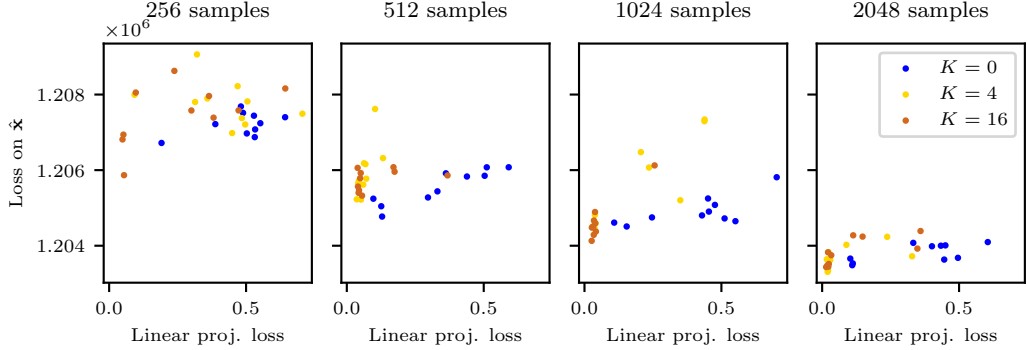

Figure 12: The effect of representation augmentation on linear projection and the reconstruction of $\hat{\mathbf{x}}$.

Additionally, we test SPS with deliberately incorrect group assumptions. The motivation is as follows. In real applications, researchers may incorrectly specify the symmetry constraint when the

data are complex or the symmetry is not known *a priori*. SPS is more useful if it works with various groups assumptions close to the truth. In the following experiment, we are surprised to find that SPS still learns interpretable representations under five out of five alternate group assumptions that we conceive by perturbing the correct group assumption.

Figure 13 shows our results with the vision task (on the bouncing ball dataset). The $x$ tick labels show the augmentation method. Its syntax is introduced in section 3.3, but just for an example here, "$(\mathbb{R}^1, +) \times SO(2)$" denotes augmenting representations by 1D translations and 2D rotations. The $y$ axis of the plot is still linear projection loss (as discussed in section 4.2.2) that evaluates the interpretability of the learned representation. As is shown by the boxplot, five out of five perturbed group assumptions yield better results than the "w/o Symmetry" baseline. Particularly, $(\mathbb{R}^3, +) \times SO(2)$ and $(\mathbb{R}^2, +) \times SO(3)$ learn significantly more linear representations, showing that some symmetry assumptions are "less incorrect" than others, and that SPS can achieve good results under a multitude of group assumptions.

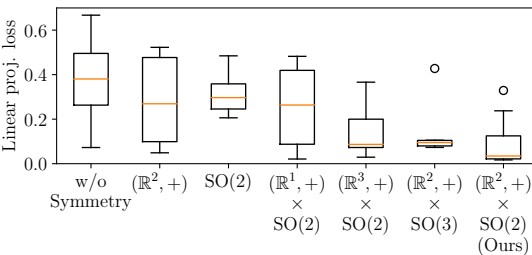

Figure 13: Evaluation on various group assumptions. The $y$ axis is linear projection loss between the learned location factor and the true coordinates, so a lower value means better interpretability of representations. The leftmost box shows the baseline without symmetry constraint. The next five boxes show five deliberately *incorrect* group assumptions. The rightmost box shows the correct group assumption.

### A.3 MORE COMPLICATED TASKS

The main part of this paper focuses on simple, straight-forward experiments. Still, we supplement our findings by reporting our current implementation's performance on more complicated tasks involving natural melody and real-world video data.

#### A.3.1 LEARNING INTERPRETABLE PITCH FACTORS FROM NATURAL MELODIES

We report the performance of SPS on learning interpretable pitch factors from monophonic melodies under a more realistic setup. We utilize the melodies from the Nottingham Dataset (Foxley, 2011), a collection of 1200 American and British folk songs. For simplicity, we quantise the MIDI melodies by eighth notes, replace rests with sustains and break down sustains into individual notes. We synthesise each non-overlapping 4-bar segment with the accordion soundfonts in FluidR3 GM (Wen, 2013), resulting in around 5000 audio clips, each of 64 steps.

This task is more realistic than the audio task described in 4.1 since we use a large set of natural melodies instead of one specified melody line. The task is also more challenging as the prior model has to predict long and more complex melodies. To account for this challenge, we use a GRU (Cho et al., 2014) with 2 layers of 512 hidden units as the prior model. We perform early-stopping after around 9000 iterations based on spectrogram reconstruction loss on the training set. The model and training setup is otherwise the same as in 4.1.

Following 4.1.2, We evaluate our approach on notes synthesised with all instruments in GeneralUser GS v1.471 (Chris, 2017) in the MIDI pitch range of C4 to C6, where most of the melodies in Foxley (2011) take place. Note that this is a challenging zero-shot scenario since the model is trained on only one instrument. We compare our model, our model ablating the symmetry loss and a $\beta$-VAE baseline. We visualise the embedded $z_{\text{pitch}}$ and synthesised pitches for different instruments in

Figure 14. Following A.2.1, $R^2$ results are shown in Figure 15 and Table 8. Even when tested on unseen timbres, our model can learn linear and interpretable pitch factors and demonstrates better embedding and synthesis performance compared with the ablation model, which outperforms the $\beta$-VAE baseline.

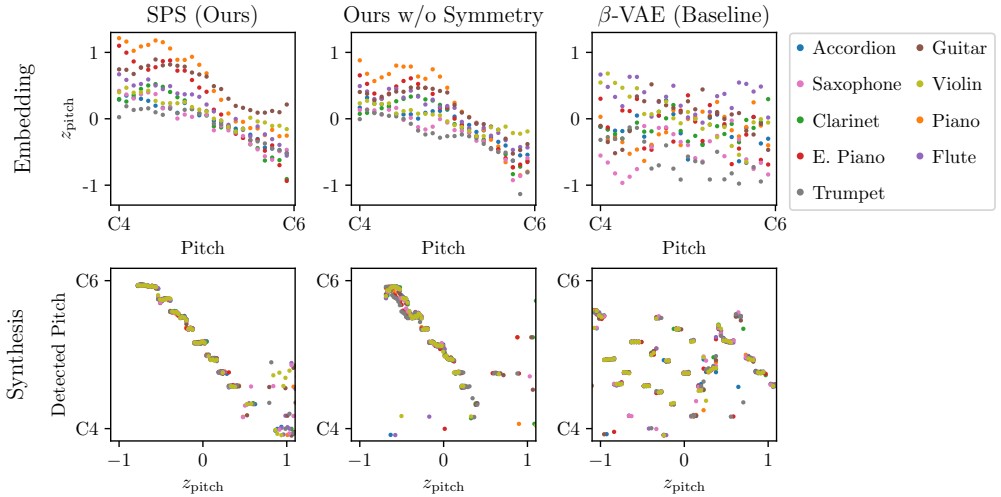

Figure 14: A visualisation of the mapping between the embedded 1D content factor and the true pitch for the model trained on Nottingham dataset.

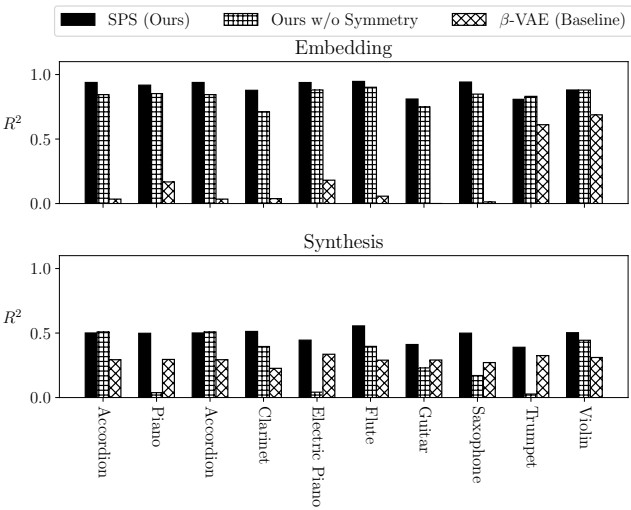

Figure 15: $R^2$ for select instruments in the test set. A larger $R^2$ indicates a more linear and interpretable latent space.

### A.3.2 LEARNING AN INTEPRETABLE LOCATION FACTOR FROM KITTI-MASKS

In this task, we evaluate our method's capability on a real-world dataset, KITTI-Masks (Klindt et al., 2021). The dataset provides three labels for each image: $X$ and $Y$ for the mask's 2D coordinate, and $AR$ for the pixel-wise area of the mask. Based on the provided labels, we use simple geometrical relation to estimate the person-to-camera distance $d$, computed as $d = 1/\tan(\alpha\sqrt{AR})$, where $\alpha$ is a constant describing the typical camera's Field of View (FoV).

We use a 3-dimensional latent code for all models. For SPS, all 3 dimensions are content factors $z_c$ and no style factor $z_s$ is used. We apply group assumption $(\mathbb{R}^3, +)$ to augment representations with

Table 8: $R^2$ aggregated across all instruments in the test set. A larger $R^2$ indicates a more interpretable latent space.

| Method | Embedding $R^2 \uparrow$ | Synthesis $R^2 \uparrow$ |
|---|---|---|
| SPS (Ours) | **0.89** | **0.47** |
| Ours w/o Symmetry | 0.83 | 0.25 |
| $\beta$-VAE (Baseline) | 0.19 | 0.29 |

$K = 1$. To measure the interpretability, we fit a linear regression from $z_c$ to the ground truth labels and calculate MSEs in the same way as in section 4.2.2. The results are shown in Table 9. Linear proj. MSE 1 measures the errors of linear regression from $z_c$ to the original dataset labels. Linear proj. MSE 2 measures the errors of linear regression from $z_c$ to the person's 3-D location, estimated from the labels.

As is shown in Table 9, MSE 2 is smaller than MSE 1 for SPS, indicating that SPS learns more fundamental factors (person's location) rather than superficial features (pixel-wise location and area). For the baseline methods, MSE 2 is almost equal to MSE 1, and both of them are higher than those of SPS. In summary, our experiment shows that SPS learns more interpretable representations than the baseline (as well as the ablation method, "Ours w/o Symmetry") on KITTI-Masks dataset.

Table 9: Results of KITTI-Masks task, averaging on 30 random initialisations for each method.

| Methods | Self-recon (BCE/Pixel) $\downarrow$ | Pred-recon (BCE/Pixel) $\downarrow$ | Linear proj. MSE 1 $\downarrow$ | Linear proj. MSE 2 $\downarrow$ |
|---|---|---|---|---|
| Ours w/o Symmetry | 0.030±0.001 | 0.093±0.010 | 0.235±0.077 | 0.243±0.088 |
| $\beta$-VAE (Baseline) | **0.028±0.001** | N/A | 0.403±0.194 | 0.399±0.204 |
| SPS (Ours) | 0.030±0.001 | **0.084±0.006** | **0.215±0.067** | **0.203±0.065** |

Figure 16 illustrates that the factors learned by SPS are more linear than those learned by other methods in the human location attribute. We choose all sequences with length $\geq 12$ from KITTI-Masks as our dataset. Specifically, we use 1058 sequences for training and 320 sequences for evaluation. In the inference stage, only the first 4 frames are given. All three methods are trained 30 times with different random initialisations. Table 9 shows the average over the 30 random initialisations, evaluated on the same test set.

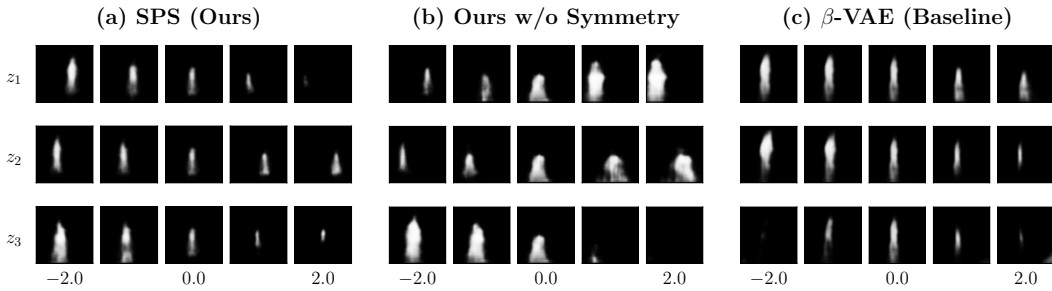

Figure 16: Latent space traversal on different models. Row $i$ shows the generated images when changing $z_i$ and keeping $z_{\neq i} = 0$. The range of $z_1$ from -2 to 2 corresponds to the human location from near-right to far-left, $z_2$ from near-left to far-right, and $z_3$ from near to far. We can see that other methods produce more non-linear trajectories, for example in (c), the human location hardly changes when $z_1 < 0$, but it changes dramatically when $z_1 > 0$.

### A.4 SPS WITH AUTOENCODER (AE)

We also test our method on AE, i.e., a model with almost the same architecture but the latent representation is not variational. Although AE usually suffers from its unregularised latent distribution and cannot learn meaningful embeddings, we find that once regularised with physical symmetry AE turns out to learn interpretable latent representations. This section presents our preliminary findings.

Table 10 supplements Table 2 by adding AE to the comparison. The latent space learned by AE is less linear than that learned by VAE but still much better than the ablation and baseline methods.

Table 10: Linear fits between the true location and the learned location factor, comparing AE with VAE.

| Method | $x$ axis MSE $\downarrow$ | $y$ axis MSE $\downarrow$ | $z$ axis MSE $\downarrow$ | MSE $\downarrow$ |
|---|---|---|---|---|
| SPS-AE | 0.17 | 0.23 | 0.24 | 0.21 |
| SPS-VAE | **0.11** | **0.06** | **0.09** | **0.09** |

We further evaluate AE's disentanglement performance following the methods in sections 4.1.3, 4.2.3 and A.2.2. Figure 17, Table 11 and Table 12 respectively supplement Figure 4, Table 1 and Table4. Figure 18, Table 13 and Table 14 supplement Figure 6, Table 3 and Table5. All results are evaluated on test sets. The AE variant of our model achieves comparable performance to the VAE in both pitch-timbre and space-colour disentanglement, even slightly surpassing VAE in several metrics. Our results show that symmetry-regularisation empowers AE to learn well-disentangled representations.

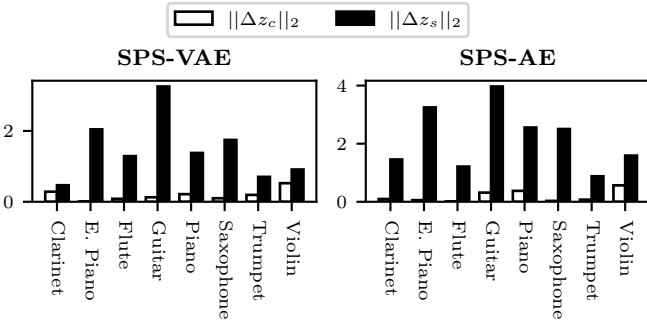

Figure 17: Comparisons for $\Delta z_c$ and $\Delta z_s$ for different instruments against accordion, with pitch kept constant at MIDI pitch D3.

Table 11: Mean ratios of $z_c$ and $z_s$ changes under timbre augmentation across all possible instrument combinations under different constant pitches. We compare the AE and VAE variants of our model.

| Methods | $||\Delta z_c||_2/||\Delta z_s||_2 \downarrow$ |
|---|---|
| SPS-AE | **0.48** |
| SPS-VAE | 0.49 |

Figure 19 is plotted in the same way as Figure 8, showing how the true location-latent space mapping becomes gradually more disentangled and linear as training progresses. AEs, once symmetry-augmented, are able to learn a location factor of similar qualities to those of VAEs within the same number of epochs.

Table 12: Results on augmentation-based queries on the audio task.

| Methods | Timbre augmentation | | Pitch augmentation | |
|---|---|---|---|---|
| | Precision ↑ | Random Precision | Precision ↑ | Random Precision |
| SPS-AE | **0.99** | 0.67 | 0.80 | 0.33 |
| SPS-VAE | 0.98 | 0.67 | **0.82** | 0.33 |

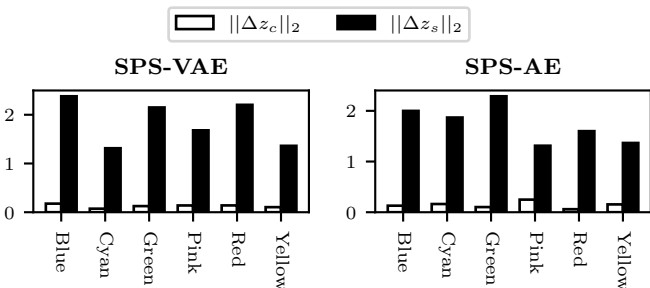

Figure 18: Comparisons for $\Delta z_c$ and $\Delta z_s$ for different colours against white, with location kept constant at (0, 1, 5).

Table 13: Mean ratios of $z_c$ and $z_s$ changes under colour augmentation across sampled colour combinations. Locations are kept unchanged.

| Methods | $||\Delta z_c||_2/||\Delta z_s||_2 \downarrow$ |
|---|---|
| SPS-AE | **0.48** |
| SPS-VAE | 0.54 |

Table 14: Results on augmentation-based queries on the visual task.

| Methods | Colour augmentation | | Location augmentation | |
|---|---|---|---|---|
| | Precision ↑ | Random Precision | Precision ↑ | Random Precision |
| SPS-AE | 0.98 | 0.40 | 0.85 | 0.40 |
| SPS-VAE | **0.99** | 0.40 | **0.88** | 0.40 |

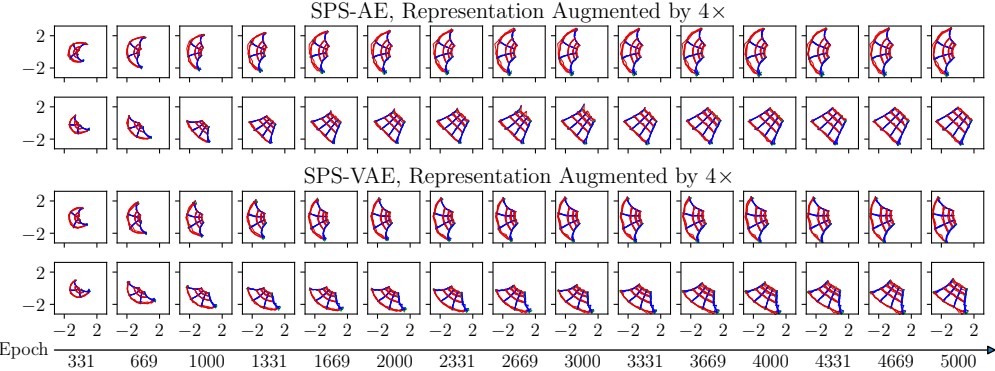

Figure 19: Comparing VAE and AE in terms of latent space regularisation during training.

Table 15: Pitch range (in MIDI note) for each instrument in our dataset.

| Instrument | MIDI Note (from) | MIDI Note (to) |
|---|---|---|
| Accordion | 58 | 96 |
| Acoustic Bass | 48 | 96 |
| Banjo | 36 | 96 |
| Baritone Saxophone | 36 | 72 |
| Bassoon | 36 | 84 |
| Celesta | 36 | 96 |
| Church Bells | 36 | 96 |
| Clarinet | 41 | 84 |
| Clavichord | 36 | 84 |
| Dulcimer | 36 | 84 |
| Electric Bass | 40 | 84 |
| Electric Guitar | 36 | 96 |
| Electric Organ | 36 | 96 |
| Electric Piano | 36 | 96 |
| English Horn | 36 | 85 |
| Flute | 48 | 96 |
| Fretless Bass | 36 | 84 |
| Glockenspiel | 36 | 96 |
| Guitar | 36 | 96 |
| Harmonica | 36 | 96 |
| Harp | 36 | 96 |
| Harpsichord | 36 | 96 |
| Horn | 36 | 96 |
| Kalimba | 36 | 96 |
| Koto | 36 | 96 |
| Mandolin | 36 | 96 |
| Marimba | 36 | 96 |
| Oboe | 36 | 96 |
| Ocarina | 36 | 96 |
| Organ | 36 | 96 |
| Pan Flute | 36 | 96 |
| Piano | 36 | 96 |
| Piccolo | 48 | 96 |
| Recorder | 36 | 96 |
| Reed Organ | 36 | 96 |
| Sampler | 36 | 96 |
| Saxophone | 36 | 84 |
| Shakuhachi | 36 | 96 |
| Shamisen | 36 | 96 |
| Shehnai | 36 | 96 |
| Sitar | 36 | 96 |
| Soprano Saxophone | 36 | 96 |
| Steel Drum | 36 | 96 |
| Timpani | 36 | 96 |
| Trombone | 36 | 96 |
| Trumpet | 36 | 96 |
| Tuba | 36 | 72 |
| Vibraphone | 36 | 96 |
| Viola | 36 | 96 |
| Violin | 36 | 96 |
| Violoncello | 36 | 96 |
| Whistle | 48 | 96 |
| Xylophone | 36 | 96 |

