# OpenReview forum: "Learning Basic Interpretable Factors from Temporal Signals via Physics Symmetry"
_ICLR.cc/2023/Conference — Submitted to ICLR 2023_

### Official Review · Reviewer_rnjU · 2022-10-24

**Confidence:** 3
**Correctness:** 3
**Technical Novelty And Significance:** 3
**Empirical Novelty And Significance:** 3
**Recommendation:** 6

**Clarity, Quality, Novelty And Reproducibility:**

The figure captions are often too short to understand what is plotted. This was mainly a problem for me in Figure 3: what exactly was the procedure to plot these two figures?  In Figure 4, the normalization by the variance or squared norm of z is necessary to make sense of the y-axis across model and within a model between content and style. It might also be useful to recall the definition of $\Delta z$ in the caption of Figure 4. I do not know what is SPS-AE in comparison with SPS-VAE in Figure 7. More information is necessary to make sense of Figure 9.

The exact definition of the random selection of times for the style vector is important: how many time steps are selected for instance?

It is not clear to me what are the instrument and timber which are used in the training set and testing sets. When reporting the sound fonts being used, more details would be necessary to know exactly which instrument (how many?), timber, loundness, are chosen, and whether the same not sequences are testing with all instruments in balanced fashion or not.

**Strength And Weaknesses:**

The paper is enjoyable to read and the idea is simple and well described.

1) I think a more precise way to describe the inductive bias of the network architecture is that it "separates some latent equivariant spatio-temporal features from time-invariant features". I think it would be more didactic to spell this out in the intro and intuition section because currently the description remains on the high level description of "physical symmetries" or "pitch vs. timbre" disentanglement which I find less precise and specific to this work.

2) Disentanglement of content versus style is an important topic, for instance to separate pitch, timbre and instrumental features. I am not convinced however that this technique has a strong reason to extract "only" the pitch with arbitrary audio because there are many other 1D features in music which can be described with a latent equivariant temporal series: loudness, note frequency, RMS ratio of high frequency vs. low frequency etc.. All these features are likely to be independent from the pitch in a real dataset and may well be what is extracting in to the latent time series z after convergence. If there is no way to prove this wrong, it might be good to report that in the discussion.

3) For the reason above, I expect that the inductive bias of the model would not be expressive enough to describe the full dataset. So I would expect a substantial loss of information in the latent representation of the auto-encoder. In fact this might even be true on the simple synthetic datasets that are considered because there are no metric report the "quality" of the reconstruction against the beta-VAE for instance. Maybe the beta-VAE focuses on other important features which are more crucial for the reconstruction of the audio than the pitch? Or for the same argument as before, the re-constructed audio is likely to have a wrong timber, loundness, etc.. if the pitch is the only time-variable latent that is captured. Reporting a metric of audio reconstruction quality would be a good start.

4) One possible flaw: the number of samples in the training sets look ridiculously small. I suspect that the model might not generalize well to the test/validation set in this setting. It is therefore crucial to state explicitly in the captions of ALL figures and tables if performance metric (or plotted data) is shown for samples from the training or testing sets.

5) It is not very standard to put the related work in discussion rather than introduction. It makes the whole idea of group symmetry sound a lot model novel that it is at first sight. I would encourage the authors to put at least a few references about disentangling AEs and group symmetry in the intro for this purpose. Also i am wondering but I am not sure, wouldn't it be fair to cite some papers from the Max welling group who have used extensively symmetry groups to generate equivariant representations? Here are two suggestions:

http://proceedings.mlr.press/v121/ilse20a.html

http://proceedings.mlr.press/v48/cohenc16.html


**Summary Of The Paper:**

The paper describes an auto-encoder setup which aims to disentangle content and style using multiple inductive biases. (1) The style given by selecting random times-slices of the encoded time series, (2) the encoded time series need to be predictable 1-step-ahead using a recurrent network R and (3) the encoded time series need to be equivariant to random translation or rotations S. The setup is tested on two synthetic datasets: a synthesized audio dataset with ascending and descending note sequences, and an short video clips of a bouncing ball. On these datasets, pitch or ball position seem to be well separated from the timber and the instrument texture.

**Summary Of The Review:**

The idea is simple and well described, but it is not clear whether this method would be capable to disentangle content and style on a real dataset because (1) it is only tested on simple synthetic datasets and (2) some simplifying 1D-latent assumptions are necessary for the network architecture to reconstruct the input faithfully.

I believe I will increase my score if my comments are addressed appropriately and I become sure that the experiments and the Figures are flawless and the paper is transparent about the computed performance metrics (is that train or test? how many instruments? normalization? why choosing beta-VAE as a baseline and then GMVAE etc...)

---

> ### Author Response · Authors · 2022-11-19
> **Reply 3 / 3**
>
> > The figure captions are often too short to understand what is plotted. This was mainly a problem for me in Figure 3: what exactly was the procedure to plot these two figures? In Figure 4, the normalization by the variance or squared norm of z is necessary to make sense of the y-axis across model and within a model between content and style. It might also be useful to recall the definition of  Δz in the caption of Figure 4. I do not know what is SPS-AE in comparison with SPS-VAE in Figure 7. More information is necessary to make sense of Figure 9.
>
> **Reply:** We have changed the caption style in the revision; please have a look. (We were writing in a style that captions alone do not reveal all the information and the interpretation, and explanations of figures are in the main text.) For reference, here we quote some of our updated captions of the figures mentioned in the review.
>
> Figure 3:
> A visualisation of the mapping between the 1D content factor and the true pitch. \blue{In the upper row, models encode notes in the test set to $z_\mathrm{pitch}$. The $x$ axis shows the true pitch and the $y$ axis shows the learned pitch factor. In the lower row, the $x$ axis traverses the $z_\mathrm{pitch}$ space. The models decode $z_\mathrm{pitch}$ to audio clips. We apply YIN to the audio clips to detect the pitch, which is shown by the $y$ axis. In both rows, a linear, noiseless mapping is ideal, and our method performs the best. All results are evaluated on the test set.
>
> Figure 4:
> Comparisons for $\Delta z_c$ and $\Delta z_s$ for different instruments against accordion, with pitch kept constant at MIDI pitch D3. $\Delta z_c$ and $\Delta z_s$ are changes in normalised $z_c$ and $z_s$, so that higher black bars relative to white bars means better results. All results are evaluated on the test set.
>
> Figure 7:
> Linear projection MSE for different augmentation factors ($K$) and training set sizes. Representation augmentation improves sample efficiency as smaller values mean better results.
>
> Figure 9:
> A visualisation of the learned latent space against training epoch. We plot how the encoder projects an equidistant 3D grid of true Cartesian coordinates onto the $z$ space. Different colours denote respective axes in the true coordinates.
>
> > The exact definition of the random selection of times for the style vector is important: how many time steps are selected for instance?
>
> **Reply:** In the training stage, one style vector is randomly selected from all time steps (i.e., 15 for the music task and 20 for the vision task) of the sequence to represent $z_s$. In the testing stage, only the first 5 (vision task) or 3 (music task) frames are given, and $z_s$ will be selected from them. We have added this in A1.3 in revision.
>
> > It is not clear to me what are the instrument and timber which are used in the training set and testing sets. When reporting the sound fonts being used, more details would be necessary to know exactly which instrument (how many?), timber, loundness, are chosen, and whether the same not sequences are testing with all instruments in balanced fashion or not.
>
> **Reply:** Thank you for your suggestion. We have added these details in our revision (section A1.3).
>
> > The idea is simple and well described, but it is not clear whether this method would be capable to disentangle content and style on a real dataset because (1) it is only tested on simple synthetic datasets and (2) some simplifying 1D-latent assumptions are necessary for the network architecture to reconstruct the input faithfully.
> I believe I will increase my score if my comments are addressed appropriately and I become sure that the experiments and the Figures are flawless and the paper is transparent about the computed performance metrics (is that train or test? how many instruments? normalization? why choosing beta-VAE as a baseline and then GMVAE etc...)
>
> **Reply:** To summarise our previous replies, we have added extra experiments with more realistic and complex datasets to assist illustrating SPS’s effect. The “1D latent assumption” may look arbitrary in specific experiments but it is in fact part of the group assumption used for the music audio task. We have also clarified various details in the revised submission. (Train or test; How many instruments; Normalisation; etc.) Also, to recap, the disentanglement is performed at two levels: a) content-style separation, and b) to further disentangle the content so that each dimension of $z_c$ is interpretable. For the music problem, GMVAE is the baseline for problem (a) and beta-vae is the baseline for problem (b).
>
> Finally, please take a look at our Common Reply: [https://openreview.net/forum?id=ifaAztwEHIN&noteId=MY7ic12MNYJ](https://openreview.net/forum?id=ifaAztwEHIN&noteId=MY7ic12MNYJ)

---

> > ### Comment · Reviewer_rnjU · 2022-11-22
> > **Thank you**
> >
> > Thank you for your reply.
> >
> > I have read you replies and I am satisfied with them, I will increase my grade to 6.
> >
> > If the authors did test their method on a real dataset (monophonic melodies as reported in the common reply) please report that clearly and rigorously in the paper: which dataset, provide audio samples and verify that the latent variable that is extracted is indeed the pitch.

---

> > > ### Author Response · Authors · 2022-12-05
> > > **Response to reviewer rnjU**
> > >
> > > Thank you for the reply. In A.3.1, we describe the monophonic melody dataset and evaluate the interpretability of the latent code our method extracts. We will also include audio samples in the revised version if the paper is accepted. We provide a few of the audio samples through this anonymous link: https://we.tl/t-poyo2Kn0Af (available until Dec 10).

---

> ### Author Response · Authors · 2022-11-19
> **Reply 2 / 3**
>
> > One possible flaw: the number of samples in the training sets look ridiculously small. I suspect that the model might not generalize well to the test/validation set in this setting. It is therefore crucial to state explicitly in the captions of ALL figures and tables if performance metric (or plotted data) is shown for samples from the training or testing sets.
>
> **Reply:** Oh, we feel flattered to see this comment. First of all, to clarify, all figures and tables show results on the testing set. (We further clarify this in the revised submission to avoid confusion.) Moreover, the model could work on “ridiculously small” training sets because of the novel training method, representation augmentation. This method dramatically boosts the sample efficiency. Please take a second look at Figure 8 in section 5 (Analysis), which shows that a “$\times 4$ ($K=4$)” representation augmentation on 512 samples (which creates 2048 total samples) leads to even better results than training on 2048 real samples! Representation augmentation guided by physical symmetry not only learns interpretable representations but also increases sample efficiency, which is a major contribution of this paper.
>
> > It is not very standard to put the related work in discussion rather than introduction. It makes the whole idea of group symmetry sound a lot model novel that it is at first sight. I would encourage the authors to put at least a few references about disentangling AEs and group symmetry in the intro for this purpose. Also i am wondering but I am not sure, wouldn't it be fair to cite some papers from the Max welling group who have used extensively symmetry groups to generate equivariant representations? Here are two suggestions:
> - http://proceedings.mlr.press/v121/ilse20a.html
> - http://proceedings.mlr.press/v48/cohenc16.html
>
>
> **Reply:** Thanks for the suggested papers, and we included them in the revised related work. We’d like to emphasise that our method of using physical symmetry is indeed very novel. Though there exist many works (including what you suggested) about “group symmetry”, the essence is totally different (only the terminology is similar). As stated in related work (also reflected in our title), the novelty of our paper is not to use group symmetry, but to apply group symmetry to the prior model, which leads to representation augmentation, a totally new training technique. Such a difference is analogous to the difference between “geometrical symmetry” and “physical symmetry”. The former is merely talking about how x (the data) and z (the representation) are related, while the latter is about the regularity of the dynamics of z. Physical symmetry puts no assumptions on x and therefore is an inductive bias that is way more domain-general.

---

> ### Author Response · Authors · 2022-11-19
> **Reply 1 / 3**
>
> We thank the reviewer for giving those insightful feedback and raising those important questions. In the below chain of comments, we respond in a breakdown format.
>
> > I think a more precise way to describe the inductive bias of the network architecture is that it "separates some latent equivariant spatio-temporal features from time-invariant features". I think it would be more didactic to spell this out in the intro and intuition section because currently the description remains on the high level description of "physical symmetries" or "pitch vs. timbre" disentanglement which I find less precise and specific to this work.
>
> **Reply:** Thank you for the suggestion. “Equivariant spatio-temporal features” is a very precise description and we have incorporated that in the revision. We’d also like to take this opportunity to emphasise that, as stated in the beginning of section 3,  the disentanglement is performed at two levels: 1) content-style separation, and  2) to further disentangle the content so that each dimension of $z_c$ is interpretable. (2) is our major contribution and it is achieved by using physical symmetry.
>
> > Disentanglement of content versus style is an important topic, for instance to separate pitch, timbre and instrumental features. I am not convinced however that this technique has a strong reason to extract "only" the pitch with arbitrary audio because there are many other 1D features in music which can be described with a latent equivariant temporal series: loudness, note frequency, RMS ratio of high frequency vs. low frequency etc.. All these features are likely to be independent from the pitch in a real dataset and may well be what is extracting into the latent time series z after convergence. If there is no way to prove this wrong, it might be good to report that in the discussion.
>
> **Reply:** This is a very nice question. The general answer is that there are certainly more features involved other than pitch and timbre. (Similarly, there are more features involved other than location and colour in the videos of bouncing balls.) The point is that what content is extracted is fully decided by what symmetry inductive bias we use (i.e., what kind of spatio-temporal equivariant we use). In the case of music audio, if the content feature is 1D and symmetric operation is merely 1D translation, then the extracted content factor is “pitch”. As pitch and timbre are the most fundamental factors of monophonic music, we focus on these two factors in this paper. Other symmetric assumptions will probably lead to other factors and we leave that part as future work. As for the detailed features you mentioned, some spectral related features such as the RMS between high and low frequency should be part of timbre.
>
> > For the reason above, I expect that the inductive bias of the model would not be expressive enough to describe the full dataset. So I would expect a substantial loss of information in the latent representation of the auto-encoder. In fact this might even be true on the simple synthetic datasets that are considered because there are no metric report the "quality" of the reconstruction against the beta-VAE for instance. Maybe the beta-VAE focuses on other important features which are more crucial for the reconstruction of the audio than the pitch? Or for the same argument as before, the re-constructed audio is likely to have a wrong timber, loundness, etc.. if the pitch is the only time-variable latent that is captured. Reporting a metric of audio reconstruction quality would be a good start.
>
> **Reply:** We included the results of reconstruction quality in the appendix, please see section A2.3 and A2.4. In general, the quality is not sacrificed. Note that in appendix A2.4, the blue point ($K=0$) is $\beta$-VAE. In this task the colour of the ball is the same across the entire dataset so there is no $z_s$.

---

### Official Review · Reviewer_5zXw · 2022-10-28

**Confidence:** 3
**Correctness:** 4
**Technical Novelty And Significance:** 3
**Empirical Novelty And Significance:** 2
**Recommendation:** 5

**Clarity, Quality, Novelty And Reproducibility:**

Clarity
- This paper is easy to read. The examples are presented well to validate the expected result.
- The mathematics are neat and described well.
- The choice of the audio input representation is important but it is not explained why the spectrogram with the linear frequency scale is used.  What if mel spectrogram or other log-frequency-scaled time-frequency representations (which has a shift-invariant property along the log-frequency axis) are used?

Quality
- The experiment validates the disentanglement in different perspectives using several evaluation metrics and visualizations
- The visual examples are convincing
- Related work are comprehensive and suitable to understand the relevant topics
- The supplementary materials provide more intuitive understanding of the results
- Minor fixes :  (page 4) "the content factor be to" --> "the content factor to be" (this appears twice)

Novelty
- The proposed model is well grounded by the physics-based principle.
- The representation augmentation also sound like a training technique

Reproducibility
- The author provides the training details, and source code


**Strength And Weaknesses:**

Strengths
- The application of physical symmetry to unsupervised representation learning is very intriguing.
- Experimental results clearly validates that the model learns the disentangled and interpretable representations as targeted.
- The representation augmentation idea sounds very novel. The experiment shows the effectiveness well.


Weaknesses
- The dataset used to validate the proposed learning model are too simple: monotonic musical instrument and monocular videos of bouncing ball. It is not clear how the model can be extended to more complicated data, for example, polyphonic and multi-instrument music and real-wold videos.
- If the model is applied to a very long time-series data, RNN would have a limitation in learning the long-term dependency. If the Transformer model is used instead, it can be plugged in the proposed model?


**Summary Of The Paper:**

This paper presents a self-supervised learning method that learns disentangled and interpretable representations based on physical symmetry from time-series data. The experiments with monotonic instrument sounds and monocular videos of bouncing ball validate the method by showing disentanglement in pitch-timbre and local-color, respectively. In addition, they suggested the representation augmentation technique that lead to lower linear projection loss and improves the disentanglement.





**Summary Of The Review:**

This paper is very interesting to read and the intuition from the fields of physics is implemented well as a model that learns disentangled and interpretable representations without any labels. Also, it suggests the novel representation augmentation technique for more effective model training. The experiment and analysis clearly show that  the model successively achieves the goal.

However, my overall impression is that this paper is in the stage of proof-of-concept. The example data are quite simple: single harmonic pitch and single ball object. The experiment focuses on decoupling the latent space into locational features (pitch in spectrogram and ball position) and the rest. It is not trivial to apply the proposed model to more complex data where the content latent features change over time in a more complicated manner (e.g., polyphonic music). The authors should clearly envision the potential of the proposed model that be applied to the complex data.

---

> ### Author Response · Authors · 2022-11-19
> **Reply 1 / 1**
>
> We thank the reviewer for giving those insightful feedback and raising those important questions. In the below chain of comments, we respond in a breakdown format.
>
> > The dataset used to validate the proposed learning model are too simple: monotonic musical instrument and monocular videos of bouncing ball. It is not clear how the model can be extended to more complicated data, for example, polyphonic and multi-instrument music and real-wold videos.
>
> **Reply:** Thank you for your suggestion. We tried more complicated datasets. For the music problem, we have reported the performance of SPS on real monophonic melodies (a more realistic setup). Our approach results in more linear $z_\mathrm{pitch}$ embeddings compared with the ablation model, which outperforms the β-VAE baseline. For details please see appendix A.3.1. For the vision problem, we try our method on a real-world dataset -- KITTI-Masks. Results show that the SPS again learns linear location factors of a moving person and the results outperform the baselines. please see appendix A.3.2.
>
> > If the model is applied to a very long time-series data, RNN would have a limitation in learning the long-term dependency. If the Transformer model is used instead, it can be plugged in the proposed model?
>
> **Reply:** Since our synthetic data have simple transition rules, we use simple prior models. We agree with the reviewer and indeed wish to test SPS’s effect on more powerful prior models. Our future work already involves modelling more complex data.
>
> > The choice of the audio input representation is important but it is not explained why the spectrogram with the linear frequency scale is used. What if mel spectrogram or other log-frequency-scaled time-frequency representations (which has a shift-invariant property along the log-frequency axis) are used?
>
> **Reply:** Indeed, most related works usually use log-frequency spectrograms as input data to models, so it may seem unusual that we use linear frequency instead. However, the rationale of using log frequency itself is to help the model learn via adding music domain knowledge (i.e. perceived pitch is linear to log frequency). We exclude that domain knowledge to make the task more nontrivial: even when we don’t tell the model that log frequency makes more sense, the results still show that our model learns linear pitch factor using physical symmetry! In other words, with physical symmetry serving as a general inductive bias, we show that the input feature engineering/choice is not that important any more.
>
> > However, my overall impression is that this paper is in the stage of proof-of-concept. The example data are quite simple: single harmonic pitch and single ball object. The experiment focuses on decoupling the latent space into locational features (pitch in spectrogram and ball position) and the rest. It is not trivial to apply the proposed model to more complex data where the content latent features change over time in a more complicated manner (e.g., polyphonic music). The authors should clearly envision the potential of the proposed model that be applied to the complex data.
>
> **Reply:**
>
> Thank you for the suggestion. In the revision, we applied our method also on two more datasets under a more realistic and complicated setup. The results can be seen in section A.3.
> Also, we would like to take this opportunity to bring up more research context and background. It’s been a major challenge to learn pitch representation from audio in an unsupervised manner (see the reference [1, 2, 3]. Previous method couldn’t do it well even with instrument labels, and we solve the problem without instrument labels. Similarly, as far as we know, our study is the first method which can learn 3D coordinates of a moving object from monocular videos. The point is, even for synthetic data, other self-supervised methods do not yet work, and the contribution of our paper is to show that both vision and audio fundamental factors can be learned under a unified framework (physical symmetry). We believe that such a contribution alone is worth the attention of our representation learning community.
>
> [1] Luo, Y. J., Cheuk, K. W., Nakano, T., Goto, M., & Herremans, D. (2020, October). Unsupervised Disentanglement of Pitch and Timbre for Isolated Musical Instrument Sounds. In ISMIR (pp. 700-707).
> [2] Gfeller, B., Frank, C., Roblek, D., Sharifi, M., Tagliasacchi, M., & Velimirović, M. (2020). SPICE: Self-supervised pitch estimation. IEEE/ACM Transactions on Audio, Speech, and Language Processing, 28, 1118-1128.
> [3] Luo, Y. J., Agres, K., & Herremans, D. (2019). Learning disentangled representations of timbre and pitch for musical instrument sounds using gaussian mixture variational autoencoders. arXiv preprint arXiv:1906.08152.
>
> Finally, please take a look at our Common Reply: [https://openreview.net/forum?id=ifaAztwEHIN&noteId=MY7ic12MNYJ](https://openreview.net/forum?id=ifaAztwEHIN&noteId=MY7ic12MNYJ)

---

### Official Review · Reviewer_AY1K · 2022-10-30

**Confidence:** 4
**Correctness:** 3
**Technical Novelty And Significance:** 3
**Empirical Novelty And Significance:** 2
**Recommendation:** 6

**Clarity, Quality, Novelty And Reproducibility:**

I was confused for longer than I should have been about why the authors do not do full 3D transformations on the video data? Consequently in Section 4.2.3 I was confused by the dicussion of how "the ablation model does not differently constrain z2 (corresponding to the y-axis)". It took me a while to figure out that the y axis is special because of gravity. :-)

Otherwise the paper is plenty clear.

The submission is high quality, although it is unclear if the claims will hold up on real data.

I do not know the literature well enough to judge the novelty of the contribution. To be honest it feels like an obvious way to enforce symmetry, it's just that these simple symmetries almost never apply. (E.g. the symmetries used here aren't present in real audio or video.)

The authors have provided PyTorch code to reproduce their experiments.

**Strength And Weaknesses:**

Strengths:
  - The approach is simple and sound.
  - It is easy to understand what is being done.
  - The experiments are convincing, if contrived (more below).
Weaknesses:
  - There is no discussion of the random pooling on the style. This random pooling enforces its own symmetry -- it encourages the style representation to be constant over time. I believe that without the random pooling, the method would not work; the model would choose to convey all its information through the unconstrained style channel and not bother with the content channel.
  - Related to the previous point, the effectiveness of the method hinges on being able to partition the latent space into subspaces and coming up with appropriate symmetries for all subspaces. The experiments are performed on synthetic data that is contrived to have this exact physical-content-plus-constant-style breakdown, and it is unclear what one would do in real-world settings where this breakdown does not make sense.
  - The experiments that study the impact of style changes in the input use norms to measure the extent of change in the latents. Norms aren't really meaningful here; a small delta norm ||dx|| may correspond to a large change if ||x|| << ||dx||, and a large delta norm ||dx|| may correspond to a small change if ||x|| >> ||dx||. Consider using relative norms ||dx||/||x|| instead.

**Summary Of The Paper:**

The authors propose a method to train sequence models that encourages disentangling physical/dynamic features ("content") from constant features ("style"). The method partitions the latent features into two subsets and enforces certain symmetries by applying appropriate random transformations and penalizing the latent transition model (an RNN) for deviations from equivariance. Experiments on synthetic data show that the content features take on straightforward interpretable meanings, and that style changes in the input do not affect the content features.

**Summary Of The Review:**

This is a solid contribution. My main criticism is the applicability of the method beyond very simple settings.

---

> ### Author Response · Authors · 2022-11-19
> **Reply 1 / 1**
>
> We thank the reviewer for giving those insightful feedback and raising those important questions. In the below chain of comments, we respond in a breakdown format.
>
> > There is no discussion of the random pooling on the style. This random pooling enforces its own symmetry -- it encourages the style representation to be constant over time. I believe that without the random pooling, the method would not work; the model would choose to convey all its information through the unconstrained style channel and not bother with the content channel.
>
> **Reply:** Not quite. As stated in the beginning of section 3 (Methodology), the disentanglement is performed at two levels: 1) disentangle content and style, and 2) further disentangle the content so that each dimension of $z_c$ is interpretable. You are basically asking that if we don’t do (1), could (2) still be learned. The answer is yes. If we don’t do random pooling on the style factor, it means we don’t perform “content-style” disentanglement. In this case, we would simply not split $z$ into $z_c$ and $z_s$, and let everything be the content factor constrained by physical symmetry. Actually, the entirety of section 5 (Analysis) shows our results in this case: physical symmetry alone learns 3D coordinates of (fixed-colour) bouncing balls.
>
> > Related to the previous point, the effectiveness of the method hinges on being able to partition the latent space into subspaces and coming up with appropriate symmetries for all subspaces. The experiments are performed on synthetic data that is contrived to have this exact physical-content-plus-constant-style breakdown, and it is unclear what one would do in real-world settings where this breakdown does not make sense.
>
> **Reply:** We have added experiments on more realistic and complicated setups. Please see Appendix A.3.
>
> > The experiments that study the impact of style changes in the input use norms to measure the extent of change in the latents. Norms aren't really meaningful here; a small delta norm ||dx|| may correspond to a large change if ||x|| << ||dx||, and a large delta norm ||dx|| may correspond to a small change if ||x|| >> ||dx||. Consider using relative norms ||dx||/||x|| instead.
>
> **Reply:** That’s a good point. That’s a good point. We actually did use the normalised $z_c$ and $z_s$ for the experiments in sections 4.1.3, 4.2.3 and A.3 to ensure they have the same dynamic range. We have further clarified this in the revised submission to avoid confusion.
>
> > I do not know the literature well enough to judge the novelty of the contribution. To be honest it feels like an obvious way to enforce symmetry, it's just that these simple symmetries almost never apply. (E.g. the symmetries used here aren't present in real audio or video.)
>
> **Reply:** We are actually very proud to see this comment. Many great methods look straightforward and obvious, but very few people could come up with such concise solutions. To bring up some contexts, it’s been a major challenge to learn pitch representation from audio in an unsupervised manner (see the reference [1, 2, 3]. Previous method couldn’t do it well even with instrument labels, and we solve the problem without instrument labels. Similarly, as far as we know, our study is the first method which can learn 3D coordinates of a moving object from monocular videos. The point is, even for synthetic data, other self-supervised methods do not yet work, and the contribution of our paper is to show that both vision and audio fundamental factors can be learned under a unified framework (physical symmetry). We believe that such a contribution alone is worth the attention of our representation learning community.
>
> To further illustrate our idea, we applied our method on two more datasets in a more realistic and complicated setup. The results can be seen in section A.3 in the revised version.
>
> [1] Luo, Y. J., Cheuk, K. W., Nakano, T., Goto, M., & Herremans, D. (2020, October). Unsupervised Disentanglement of Pitch and Timbre for Isolated Musical Instrument Sounds. In ISMIR (pp. 700-707).
> [2] Gfeller, B., Frank, C., Roblek, D., Sharifi, M., Tagliasacchi, M., & Velimirović, M. (2020). SPICE: Self-supervised pitch estimation. IEEE/ACM Transactions on Audio, Speech, and Language Processing, 28, 1118-1128.
> [3] Luo, Y. J., Agres, K., & Herremans, D. (2019). Learning disentangled representations of timbre and pitch for musical instrument sounds using gaussian mixture variational autoencoders. arXiv preprint arXiv:1906.08152.
>
> Finally, please take a look at our Common Reply: [https://openreview.net/forum?id=ifaAztwEHIN&noteId=MY7ic12MNYJ](https://openreview.net/forum?id=ifaAztwEHIN&noteId=MY7ic12MNYJ)

---

> > ### Comment · Reviewer_AY1K · 2022-11-24
> > **A few more questions**
> >
> > Thank you for your responses. I'm not entirely convinced that your method is better than data augmentation when all things are considered.
> > Your argument that data augmentation requires *more* domain knowledge is very compelling.
> > On the other hand, data augmentation is much less invasive, because it does not require a tiny latent space like you have here (2 units for the audio task, 4 units for the bouncing ball task).
> > Your latent space is tiny because you have to manually decompose it into parts and impose plausible constraints on each part.
> > How are you going to do this if you have a hundred latent dimensions?
> > In your updated draft you show some resilience to inappropriate choices of symmetry, but I don't think this is representative of what will happen in a case like this.
> > Typically with neural networks, we just choose some large number of latent units and trust that it's large enough.
> > Having to justify each and every latent dimension with domain knowledge is a huge price to pay.
> >
> > Also, I'd like to respond to this specifically:
> >
> > > You are basically asking that if we don’t do (1), could (2) still be learned. The answer is yes. If we don’t do random pooling on the style factor, it means we don’t perform “content-style” disentanglement. In this case, we would simply not split $z$ into $z_c$ and $z_s$, and let everything be the content factor constrained by physical symmetry. Actually, the entirety of section 5 (Analysis) shows our results in this case: physical symmetry alone learns 3D coordinates of (fixed-colour) bouncing balls.
> >
> > That's not what I'm asking.
> > What I meant is to let $z_s$ be unconstrained, neither by pooling nor by symmetry.
> > Generally, if there are two paths for information to flow through ($z_s$ and $z_c$), a neural network will choose to cram everything through the unconstrained path.
> > So what I was asking is: say you have a hundred latent units for your bouncing ball videos, can you restrict three of them to be physical and leave the rest to encode whatever else?
> > If you could, this would make your method applicable to natural video, in which some things some times benefit from being represented in equivariant ways, but not everything all the time.
> > For example, video of soccer matches, in which there are times when it makes sense to represent aspects of the ball and players physically, while at the same time there are other phenomena (moving cameras, scene cuts) for which the appropriate symmetries are not obvious.
> >
> > I disagree that your experiment in Section 5 addresses the issue.
> > Yes, you're no longer doing content-style disentanglement, but you've also removed the style variation from your data by choosing to train only on green balls.
> > As a result, the style information does not have to be conveyed through the latent representation, and can just be assumed by the decoder.
> > If you trained this variant on all three colors, it might work, but *in spite* of the inappropriate physical symmetry imposed on the style channel.
> > How would you apply your approach on videos where the ball's color is not constant, but changes abruptly and unpredictably?
> > In this case temporal pooling for style would be inappropriate.
> >
> > To be clear, I'm not demanding you demonstrate your method on a big real-world problem; that is not necessary for a good paper. My issue with the paper right now it is not clear *even in principle* how you would apply it to such a problem.

---

> > > ### Author Response · Authors · 2022-12-02
> > > **Response to reviewer AY1K**
> > >
> > > Thank you for your comments and follow-up questions.
> > >
> > > To clarify, we are certainly *not* saying that representation augmentation is a better option than data augmentation in all cases. These two have different advantages in different scenarios. For *interpretable* representation learning tasks considered in this paper, representation augmentation can achieve something data augmentation cannot, and we see this novel method as a major contribution of this paper. To be specific, we know that data augmentation can improve model robustness in terms of reconstruction and prediction, but it *cannot* learn a linear pitch concept from music audio (or learn 3d cartesian coordinates from video of a simple moving object) as what representation augmentation could do. Indeed, if we only care about reconstruction and prediction at the data level, using a large number of hidden units is practical. However, if we truly care about the *interpretability* (which is the focus of our paper) of the latent representation, *the unit number has to be shrunk dramatically*. Actually, it’s our belief that human minds perceive complex scenes using extremely compact latent space, and future AI models (if we care about interpretability) should behave in a similar way. We see our method as an important step towards this goal.
> > >
> > > As you pointed out, “to specify each and every latent dimension with domain knowledge is a huge price to pay”, and that’s exactly the bottleneck of existing methods with lots of hidden units. We are seeking another path: low-dim latent space whose interpretability is achieved via equivariance and global invariance constraints, and the learning technique (representation augmentation) doesn’t require much domain knowledge as in data augmentation.
> > >
> > > Being that said,  you are absolutely correct that if we add many redundant free dimensions to the latent space and only constrain a small number of dimensions by physical symmetry, our method will not work. But we will not do such a thing since in this paper we mainly consider simple cases where features follow symmetry and global invariance alone can reconstruct the data. Actually, the experiment on KittiMasks (Table 9) shows that if there are extra features but we only assume symmetric ones, the reconstruction accuracy is sacrificed but interpretability is still better than the baseline.
> > >
> > > Above all, this is a very useful and insightful discussion, and we realise that the true difference between complicated and simple scenarios is whether *symmetric features alone are capable of reconstruction*. We agree with your argument that in much more complicated setups, it is ideal to assume some free dimensions that are *not* interpretable but useful for reconstruction. In principle, this can be achieved by enforcing extra cycle consistency in our symmetric framework, and we leave it for future work.

---

### Official Review · Reviewer_9KCD · 2022-11-02

**Confidence:** 3
**Correctness:** 2
**Technical Novelty And Significance:** 3
**Empirical Novelty And Significance:** 3
**Recommendation:** 5

**Clarity, Quality, Novelty And Reproducibility:**

The use of English is good.  I did not find any spelling or grammatical inaccuracies.

The concepts proposed and designs chosen in this paper are not well motivated in the text.  A reader can possibly guess why the authors made each proposal and design choice, but it would be better to not leave the reader guessing.

The paper proposes two novel forms of inductive biases that can be used, with the aim of not needing domain expertise.  However, the proposal does require domain expertise to choose S.

The data and modelling approached used in the experiments should be reproducible.  A Github repository exists for this paper.


**Strength And Weaknesses:**

Strengths:
- It is not trivial to experimentally assess whether the proposal is doing what the authors claim that it should be doing.  The authors go through a lot of effort to run many different types of innovative tests to assess this behaviour.

Weaknesses relating to unclear description of the proposed approach:
- How do you split z into z_s and z_c?
- Why does transforming z through S yield a fake sequence?  What does "fake sequence" mean in this context?
- The model described in figure 2 does not have any K.  Yet, section 3.3 mentions K different transformations.  Are the descriptions consistent?
- In the beta-VAE baseline in figure 3, how is the pitch estimated from the multi-dimensional embeddings?
- What is a "bouncing force"?

Weaknesses relating to lack of motivation for why design choices were made:
- What is the motivation for using random pooling in P, as opposed to mean pooling or self-attention?
- What is the motivation for having each of the three branches in the model?
- In the introduction, the proposal is motivated as being better than previous works, because there is no need for domain expertise.  However, S needs to be chosen with domain expertise.  Is the original motivation still satisfied?
- What is the motivation and mathematical definition of each term in the training criterion?  "BCE" is not defined in the paper.  l_2 is not defined.  Why does the KL divergence measure a distance between z_i and a standard Gaussian?

Weaknesses relating to ambiguity in the language used:
- What does "group assumption" mean?

Weaknesses relating to questionable design choices:
- It is claimed that the proposed inductive bias separates out pitch and timbre, or location and colour.  How do you know that these are the only factors involved?  What about other factors, such as volume, speed, starting location, etc.?
- In section 4.1.3, how do you know that the difference in magnitude of delta z_c and delta z_s is not because z_c and z_s learn to have different dynamic ranges?  Perhaps delta z_c and delta z_s should first be normalised by the dynamic ranges of z_c and z_s.
- In section 4.2.2, why do the movement directions conveniently align with the standard basis of z?  Why do they not align with an arbitrary rotation of the standard basis of z?  Maybe the movement directions are also aligned with an arbitrary rotation of the z basis for beta-VAE?


**Summary Of The Paper:**

This paper proposes novel ways to enforce an inductive bias into a temporal auto-encoder model, by:
1) enforcing that the R part of the model should be able to operate on both the original version of the time-varying embeddings and also on a transformed version.
2) separating out the per-session and per-frame parts of the embeddings.

The motivation of enforcing 1. is to allow for an inductive bias that does not require domain expertise to design.  However, the proposal requires domain expertise to choose the S transformation, which goes against the original motivation.


**Summary Of The Review:**

The proposal is motivated as being able to avoid domain expertise, but in the end domain expertise is needed to choose S.
The design choices are not well motivated in the text.

Ithenticate similarity is 3%, which is good.

---

> ### Author Response · Authors · 2022-11-19
> **Reply 3 / 3**
>
> > In section 4.1.3, how do you know that the difference in magnitude of delta z_c and delta z_s is not because z_c and z_s learn to have different dynamic ranges? Perhaps delta z_c and delta z_s should first be normalised by the dynamic ranges of z_c and z_s.
>
> **Reply:** As stated in section 4.1.3,  we did use the normalised $z_c$ and $z_s$ for the experiments in sections 4.1.3, 4.2.2, 4.2.3 and A.3 to ensure they have the same dynamic range. We further clarify this in the revised submission to avoid confusion.
>
> > In section 4.2.2, why do the movement directions conveniently align with the standard basis of z? Why do they not align with an arbitrary rotation of the standard basis of z? Maybe the movement directions are also aligned with an arbitrary rotation of the z basis for beta-VAE?
>
> **Reply:** If you look carefully at figure 6, the movement directions do not align exactly with the standard basis of $z$. Only the vertical movement alway aligns with the vertical standard basis of $z$, and the movements on the horizontal plane align with an arbitrary rotation of the other two standard bases of $z$. This is due to the group assumption we use (A1.2) and the ball's vertical movement is special (because of gravity and bouncing). Obviously, the movement directions for beta-VAE are not aligned with any linear transformation of the z basis, which can be seen from Figure 6. This can also be seen in table 2. To avoid those confusion, we will also change the main text discussing Figure 6 in the revised submission.
>
> Finally, please take a look at our Common Reply: [https://openreview.net/forum?id=ifaAztwEHIN&noteId=MY7ic12MNYJ](https://openreview.net/forum?id=ifaAztwEHIN&noteId=MY7ic12MNYJ)

---

> ### Author Response · Authors · 2022-11-19
> **Reply 2 / 3**
>
> > In the introduction, the proposal is motivated as being better than previous works, because there is no need for domain expertise. However, S needs to be chosen with domain expertise. Is the original motivation still satisfied?
>
> **Reply:** We respect physical symmetry as a general inductive bias since it is ubiquitous in a variety of temporal processes including vision and audio tasks. We do need to define the symmetry constraints for each specific task such as translation and rotation, which, however, are very easy to find intuitively. To compare with previous works (say, learning pitch representations) our method does not require any particular knowledge on the training spectrogram (i.e., the timbre label or the log-linear mapping between frequency and pitch). Of course, we still need some inductive bias, but the inductive bias used in our method (physical symmetry) is much more general.
>
> Please also refer to our Common Reply, where we elaborate on this issue: [https://openreview.net/forum?id=ifaAztwEHIN&noteId=MY7ic12MNYJ](https://openreview.net/forum?id=ifaAztwEHIN&noteId=MY7ic12MNYJ)
>
> > What is the motivation and mathematical definition of each term in the training criterion? "BCE" is not defined in the paper. l_2 is not defined. Why does the KL divergence measure a distance between z_i and a standard Gaussian?
>
> **Reply:** BCE is Cross Entropy Loss and l_2 is mean-squared loss, both common to modern works in deep learning. We use BCE since it measures the distance between reconstructed image and ground truth image. l_2 loss measures the distance between two $z$ sequences. In VAEs, $z$ is regularised by a standard gaussian prior, and In VAEs, $z$ is regularised by a standard gaussian prior, and KL divergence measures how far apart the two distributions (the standard gaussian prior and the posterior distribution of $z$) are. The KL divergence of two gaussians can be easily performed, and it measures how far apart the two distributions (the standard gaussian prior and the posterior distribution of $z$) are. The KL divergence of two gaussians can be easily performed, and it is a standard practice in VAE training.
>
> > What does "group assumption" mean?
>
> **Reply:** Group assumption is the assumption about what  group (mathematics)roup (mathematics) is used for representation augmentation.  used for representation augmentation. We first mention group assumption in section 3.3 and we write that it is what an imaginary experience is based on, so group assumption is the operation $S$ in this context.
>
> > It is claimed that the proposed inductive bias separates out pitch and timbre, or location and colour. How do you know that these are the only factors involved? What about other factors, such as volume, speed, starting location, etc.?
>
> **Reply:** There are certainly other factors involved in the data, but our model learns equivariant content factors (with respect to the physical law) and the global invariant style factor. The content and style factors, which are fundamental factors, turn out to be pitch and timbre for the music problem and location and colour for the video problem. You can regard our group assumption as constraints so that only factors conform to this assumption are learned as latent codes. Learning other factors violates the constraints and results in higher symmetry-based losses, so they have to be calculated in other ways.

---

> ### Author Response · Authors · 2022-11-19
> **Reply 1 / 3**
>
> We thank the reviewer for giving those insightful feedback and raising those important questions. In the below chain of comments, we respond in a breakdown format.
>
> > How do you split z into z_s and z_c?
>
> **Reply:** To split $z$ means to cut the vector $z$ into two parts. If $z$ has $n$ dimensions , we choose $z[0:m]$ as $z_c$ and $z[m:n]$ as $z_s$, where $n$ and $m$ are hyperparameters (different numbers are used for different problems). We have further clarified this in appendix A.1.2 in the revised submission to avoid confusion.
>
>
> > Why does transforming z through S yield a fake sequence? What does "fake sequence" mean in this context?
>
> **Reply:** A “fake sequence” (of $z$) is a sequence that is imaginary and doesn’t originally exist in the training data. Just as “data augmentation creates “fake” data samples, our representation augmentation technique creates “fake” representations, denoted as $z^{S}$. In section 2, we define representation augmentation as transforming $z$ through $S$, and explain that representation augmentation entails ''imagined'' new $z^{S}_{t}$ serving as the training samples for the prior model.
>
> > The model described in figure 2 does not have any K. Yet, section 3.3 mentions K different transformations. Are the descriptions consistent?
>
> **Reply:** Yes, the descriptions are consistent. $K$ is a hyperparameter of our proposed training technique, representation augmentation. Similar to how papers don’t usually show data augmentation in the model architecture, we don’t show representation augmentation and its corresponding hyperparameters in figure 2. We mention $K$ only when we need to explain the details of representation augmentation, for example in section 3.3.
>
> > In the beta-VAE baseline in figure 3, how is the pitch estimated from the multi-dimensional embeddings?
>
> **Reply:** The β-VAE baseline is trained to reconstruct single-note spectrograms of a single instrument, and we set its latent dimension to 1.
>
> The beta-VAE is trained on a simpler task: to reconstruct timbre-invariant pitch-varying audio with a one-neuron bottleneck. Note there is a lossless, reversible mapping between the audio and the pitch. We use that as an ideal environment for a β-VAE to learn a pitch factor (in favour of the baseline). To clarify that point, we have updated section 4.1.2 in the paper:
> 3) a $\beta$-VAE trained to encode single-note spectrograms from a single instrument (banjo) to 1D embeddings.
>
> > What is a "bouncing force"?
>
> **Reply:** Bouncing force describes this phenomenon: As the ball is elastic, when it hits the ground, it loses a constant percentage of kinetic energy and then bounces up. The bouncing force is sometimes called the elastic force or the normal force.
>
> > What is the motivation for using random pooling in P, as opposed to mean pooling or self-attention?
>
> **Reply:** Our motivation is to simply disentangle content and style representation so that we can focus on further disentangling content representation using physical symmetry (which is the main contribution in this work). As random pooling simply works, we did not try other methods. Mean pooling and self-attention should also work. Thank you for your suggestion, and we will include them in our future research.
>
> > What is the motivation for having each of the three branches in the model?
>
> **Reply:** The motivation of the green (upper) branch is to ensure the decoder estimates a reverse function of the encoder and both of them will only deal with the current frame and not participate in prediction. Then we can use the red (middle) branch to train the prior model (RNN) so that it can be fully responsible for the representation prediction. The blue (lower) branch augments the content part of the $z$ (i.e., $z_c$) by symmetric transformations $S$, requiring the transformed $z_c$ sequence to also be predicted by the prior model and be reconstructed after corresponding reverse symmetric transformations $S^{-1}$. In other words, they serve as constraints to the encoder.

---

### Author Response · Authors · 2022-11-19
**Common reply (Part 2 / 2)**

## Question B
Our datasets are toy datasets. Are the tasks even hard? Are the experiment results surprising? Will SPS work on complex data / real-world datasets?

### B.1 Even the toy tasks are otherwise almost impossible.
In the computer music domain, we know how hard it is to unsupervisedly 1) disentangle pitch and timber, and 2) learn a linear pitch concept. As far as we know, all prior works  incorporate strong domain knowledge, such as pitch shifting for pseudo-label generation [1, 2] or using instrument labels [3]. Our method is the first to solve the two problems in a self-supervised manner with domain-general inductive bias.

Likewise, learning concepts such as 3D coordinates from unlabelled video has long been a far-fetched fantasy for CV researchers. To be fair, in recent years we have seen exciting progress on this front. For example, LEAP [4] performs physically meaningful representation disentanglement via causal discovery. But even LEAP *requires* not only independent noises but also a sufficient causal structure (e.g., five or more balls interacting in the same scene via springs forces) in order to learn disentangled location factors. Our task, with only one single ball, does not have enough causal structure for LEAP to identify disentangled axes.

Our SPS method solves both of these “toy” tasks with one simplistic method. We believe that such a contribution alone is worth the attention of our representation learning community.

[1] Luo, Y. J., Cheuk, K. W., Nakano, T., Goto, M., & Herremans, D. (2020, October). Unsupervised Disentanglement of Pitch and Timbre for Isolated Musical Instrument Sounds. In ISMIR (pp. 700-707).
[2] Gfeller, B., Frank, C., Roblek, D., Sharifi, M., Tagliasacchi, M., & Velimirović, M. (2020). SPICE: Self-supervised pitch estimation. IEEE/ACM Transactions on Audio, Speech, and Language Processing, 28, 1118-1128.
[3] Luo, Y. J., Agres, K., & Herremans, D. (2019). Learning disentangled representations of timbre and pitch for musical instrument sounds using gaussian mixture variational autoencoders. arXiv preprint arXiv:1906.08152.
[4] Yao, W., Sun, Y., Ho, A., Sun, C., & Zhang, K. (2021). Learning Temporally Causal Latent Processes from General Temporal Data. arXiv preprint arXiv:2110.05428.

### B.2. We tried more complicated datasets.

For the music problem, we have reported the performance of SPS on real monophonic melodies (a more realistic setup). Our approach results in more linear $z_\mathrm{pitch}$ embeddings compared with the ablation model, which outperforms the β-VAE baseline. For details please see appendix A.3.1.
For the vision problem, we try our method on a real-world dataset, KITTI-Masks. Results show that the SPS again learns linear location factors of a moving person and the results outperform the baselines. please see appendix A.3.2.

---

### Author Response · Authors · 2022-11-19
**Common reply (Part 1 / 2)**

We thank the four reviewers for the valuable comments. In addition to the individual replies, we would like to address several common questions and concerns here. In general, the reviews raise two common, important questions:

A. In order to know what kind of symmetry constraints suit a given problem, do we need domain knowledge after all?
B. Our experiments only involve *simple* synthetic data, so what is the significance of SPS? Will SPS perform as well on real data?

## Question A
In our vision experiment (with the bouncing ball dataset), we specify a 2D translation-rotation equivariance (to learn the Cartesian coordinate factor). In our music experiment (with the major scale dataset), we specify a 1D translation equivariance (to learn the pitch factor). It seems that the augmentation (i.e., the symmetry constraint) needs to be known in advance. Isn’t that domain knowledge?

### A.1 Representation augmentation requires much less knowledge than data augmentation.
Consider pitch extraction. Almost all related works use pitch-shift augmentation, but on the data and not on the representation. It requires the knowledge of how to transform the input data to shift the pitch. Simple techniques include time stretching and spectrogram frequency stretching, and advanced techniques (vocoders etc.) involve shifting harmonics without changing the timbre. In comparison, SPS just translates the representation vector.

As for the vision task, it is not clear at all how to apply 3D spatial translation-rotation augmentation to unlabelled video. With our SPS method, one only needs to translate and rotate a representation vector of length 3.

### A.2 Domain knowledge and representation symmetry belongs to two different conceptual levels.
When we pick group assumptions for a given problem, the kind of knowledge required is more abstract (and general) than what we usually call “domain knowledge”. It is us human’s beliefs about concepts themselves, not how concepts are encoded from data.

Concretely, recall that music and vision are two vastly different fields. Their respective domain knowledge systems look nothing like each other. In comparison, SPS solves one task with 1D translation and the other with 2D translation-rotation. Their forms look overarchingly similar! To generalise, drastically different systems can share formally similar physical symmetries. The “expertise” involved in picking symmetries is utterly different from what “domain expertise” refers to.

### A.3 We may not even need a correct symmetry assumption for successful representation learning with SPS.
The introduction section of our paper mentions that many modern physicists *start* with symmetry (in law) assumptions *and then* obtain testable theories of physical laws. Notice: first there are guesses about symmetry, and after that, “domain knowledge” is created.

What it means is that symmetry is so fundamentally general that proposing symmetry assumptions is an efficient way of regularising concepts, even before we settle down with any domain knowledge. One is free to try multiple symmetry assumptions and see which ones learn better representations.

In fact, our other experiments (not reported in the paper submitted for review, but newly added into the revision) already show that even with *incorrect symmetry assumptions* SPS can still learn to extract 3D Cartesian coordinates from the unlabelled bouncing ball dataset.

Please take a look at Figure 13 in the revised submission. It shows our results with the vision task (on the bouncing ball dataset). The $x$ tick labels show the augmentation method. Its syntax is introduced in section 3.3, but just for an example here, “$(\mathbb{R}^1, +) \times \mathrm{SO}(2)$” denotes augmenting representations by 1D translations and 2D rotations. The $y$ axis of the plot is still linear projection loss (as discussed in section 4.2.2) that evaluates the interpretability of the learned representation. As is shown by the boxplot, five out of five perturbed group assumptions yield better results than the “w/o Symmetry” baseline. Particularly, $(\mathbb{R}^3, +) \times \mathrm{SO}(2)$ and $(\mathbb{R}^2, +) \times \mathrm{SO}(3)$ learn significantly more linear representations, showing that some symmetry assumptions are ``less incorrect'' than others, and that SPS can achieve good results under a multitude of group assumptions.

Those results are now reported in appendix A.2.4 of the paper.

---

### Decision · Program_Chairs · 2023-01-20

**Decision:**

Reject

**Justification For Why Not Higher Score:**

Overall we all found this paper very intriguing. However, the question around the general applicability of the proposed approach makes me wonder if this paper is ripe for publication yet. The authors argue that asking for general applicability "is like requiring the first paper of CNN to work on face recognition", which is fair. But the current paper is like the first paper of CNN but can only tell the difference between two digits on MNIST and more importantly it's not clear how to make it recognize more than that. I hope the authors can take the engaging feedback from the reviewers into account to further improve the work.

**Justification For Why Not Lower Score:**

N/A

**Metareview: Summary, Strengths And Weaknesses:**

Summary: this paper presents an encoder-decoder model where physics symmetry is used as an inductive bias to constrain the latent space for time series data (music, video). This constraint also leads to representation augmentation, which is to imagine fake latents and encourage them to be equivariant wrt certain transformations, and it helps improve sample efficiency. In the experiments, the authors show that the proposed model can linear pitch factors, as well as pitch-timbre disentanglement from unlabelled monophonic music audio. In addition, when applied to simple video data, the model is able to learn 3D Cartesian space as well as space-color disentanglement.

Strengths: All the reviewers agree the proposed method is simple and sound. The idea of applying physics symmetry to the prior model is intriguing and the experimental results are convincing (if a bit simplistic).

Weaknesses: Some of the minor weaknesses (clarity, experiment on very simple toy datasets) were addressed during the discussion phase where a lot of engaging conversations have happened. One common criticism among all the reviewers is around the general applicability of the proposed approach. There are mainly two concerns:

* Initially the authors only experimented on very simple toy datasets. After the first round of reviews, the authors responded by trying the proposed method on slightly more realistic (albeit still very preliminary) datasets. It is on one hand interesting to see representation augmentation can help with sample efficiency by allowing us to train models with such a small dataset, but on the other hand, not clear how generalizable the proposed approach can be.

* Related to the above point, some reviewers raised the question that it is not clear if the proposed approach would even work *in principle* in some more complicated settings, which is more concerning than not work *in practice* for now. There are mainly two aspects: 1) In order for the physics symmetry constraint to make sense, the model can only have a very small latent space (in the paper the latent space is often less than 5-dimensional), to which the authors argue that this could actually be a feature of the model. I am sure there will be cases where a 5-dimensional latent space would suffice, but I would also imagine these cases are not going to be very common in the real-world. 2) A possible workaround for scaling up is to increase the latent space, but only apply symmetry constraint on a small subset of the dimensions. However, both the reviewer and the authors agree the information will likely just flow through the unconstrained part of the latent space, making the constraint meaningless.